

# Laminar-turbulent transition characteristics of a 3-D wind turbine rotor blade based on experiments and computations

Özge Sinem Özçakmak[1], Helge Aagaard Madsen[1], Niels Nørmark Sørensen[1], and Jens Nørkær Sørensen[2]

[1]DTU Wind Energy, Technical University of Denmark, Aerodynamic Design, Frederiksborgvej 399, 4000 Roskilde, Denmark
[2]DTU Wind Energy, Fluid Mechanics, 2800 Lyngby, Denmark

**Correspondence:** Özge Sinem Özçakmak (ozsi@dtu.dk)

**Abstract.** Laminar-turbulent transition behaviour of a wind turbine blade section is investigated in this study by means of field experiments and 3-D computational fluid dynamics (CFD) rotor simulations. The power spectral density (PSD) integrals of the pressure fluctuations obtained from the high frequency microphones mounted on a blade section are analyzed to detect laminar-turbulent transition locations from the experiments. The atmospheric boundary layer (ABL) velocities and the turbu-

lence intensities (T.I.) measured from the field experiments are used to create several inflow scenarios for the CFD simulations. Results from the natural and the bypass transition models of the in-house CFD EllipSys code are compared with the experiments. It is seen that the bypass transition model results fit well with experiments at the azimuthal positions where the turbine is under wake and high turbulence, while the results from other cases show agreement with the natural transition model. Furthermore, the influence of inflow turbulence, wake of an upstream turbine and angle of attack (AOA) on the transition behaviour is

investigated through the field experiments. On the pressure side of the blade section, at high AOA values and wake conditions, variation of the transition location covers up to $44\%$ of the chord during one revolution, while for the no wake cases and lower AOA values, variation occurs along a region that covers only $5\%$ of the chord. The effect of the inflow turbulence on the effective angle of attack as well as its direct effect on transition is observed. Transition locations for the wind tunnel conditions and field experiments are compared together with 2D and 3D CFD simulations. In contrast to the suction side, significant

difference in the transition locations is observed between wind tunnel and field experiments on the pressure side for the same airfoil geometry. It is seen that the natural and bypass transition models of EllipSys3D can be used for transition prediction of a wind turbine blade section for high Reynolds number flows by applying various inflow scenarios separately to cover the whole range of atmospheric occurrences.

## 1   Introduction

As the wind turbine technology develops and the size of modern wind turbines grows steadily, the design process becomes highly dependent on the availability of accurate aerodynamic prediction tools. The power output and the loads on the blade are effected by the aerodynamic characteristics such as lift to drag ratio. It is known that the skin friction drag of a turbulent





boundary layer is much higher than the one of a laminar boundary layer, which makes it critical to identify which part of the flow acting on the surface is laminar, transitional or turbulent.

The most common approach in current aerodynamic prediction tools is to use fully turbulent models for the entire boundary layer on blades/airfoils, ignoring the transitional process (Sørensen, 2009). This causes an incorrect prediction of the lift and

drag forces and the stall angles. Moreover, it is seen that the transition on a wind turbine airfoil occurs over a substantial part, varying between 0% and 30%, of the chord (Özçakmak et al., 2019). Therefore, accurate prediction of the laminar-turbulent transition process is critical for design and prediction tools to be used in the industrial design process, particularly for the high Reynolds numbers experienced by modern wind turbines.

For many years, a large amount of experimental, numerical and theoretical studies have been devoted to laminar-turbulent

transition, with the low drag laminar region being separated from the turbulent region where drag increases dramatically(Arnal et al., 1998). The transition process starts with receptivity of the disturbances like Tollmien-Schlichting (T-S) waves triggered in the laminar boundary layer, followed by a linear and nonlinear instabilities ending up with turbulent spots leading eventually to turbulent flow. In linear stability theory, it is assumed that the free stream turbulence and other disturbances are small. On the other hand, it is seen that in the existence of large external disturbances, linear disturbance growth can be bypassed,

this type of transition is defined as 'bypass transition' (Morkovin, 1985). Through years, both empirical correlations (Michel (1951) and Eppler (1978)) and semi-empirical methods are developed by Dini et al. (1992) for the prediction of the boundary layer transition. Semi-empirical transition detection of incompressible 2-D boundary layers by the $e^N$ method based on linear stability theory was introduced by Van Ingen (1956) and independently by Smith and Gamberoni (1956). Since then, different versions were developed and databases are generated for more complex problems. For instance, a database method is developed

based on the stability diagrams calculated by Arnal for Hartree/Stewartson solutions of Falkner-Skan equation. Wazzan et al. (1968) and Kümmerer (1973) published improved stability calculations for attached Falkner-Skan velocity profiles. Envelope methods (Gleyzes et al., 1983) (Drela and Giles, 1987) and approximate methods (Stock and Degenhart, 1989) were developed. The effect of disturbance environment on the transition process has been studied by Morkovin (1969) and Reshotko (1969).

The transition prediction models implemented in Navier-Stokes solvers can be categorized into algebraic/integral models

and transport models (Davis et al., 2005). The first category includes empirical models, approaches based on stability theory using Orr-Sommerfeld equations and simplified $e^N$ stability models. These models requires boundary layer information that can be obtained by integration of the boundary layer quantities or using velocity profile databases to solve integral boundary layer equations. The second category includes transition models based on the solution of the transport equations, such as the $\gamma - Re_\theta$ (Langtry, 2006) (Menter et al., 2006) model, and three equation model (Walters and Leylek, 2004). Transition can be

predicted within Direct Numerical simulation (DNS) or Large eddy simulations(LES) despite the fact that cost of the simulations increases rapidly with Reynolds number (Diakakis et al., 2019). PSE (Parabolized Stability Equations) that can include non-parallel and non-linear effects are also used for the stability analysis, having less resource requirements compared to DNS (Herbert, 1997). There are also hybrid approaches such as DES (Detached Eddy Simulation) that are used with transition models. The transition phenomenon itself is a highly non-linear problem, but with a semi-empirical extension, the $e^N$ method is

commonly used since it can predict the transition position accurately. For the industrial applications, the $e^N$ method together





with empirical criteria for transition mechanisms that are not covered by this approach, such as bypass and attachment line transition, keeps its place as a practical method (Krumbein, 2009). The design process for the wings and airfoils still requires the use of laminar-turbulent transition modelling in Reynolds-averaged Navier-Stokes (RANS) solvers. The current analysis involves a coupling of the $e^N$ transition model with the RANS solver to benefit its accuracy for high Reynolds number flows

in wind turbine applications (Sørensen et al., 2014).

The experimental studies on laminar-turbulent transition on aerospace applications goes back a long way compared to the research conducted on wind turbines. While the inflow turbulence intensity for an airplane wing in cruise is lower than the one experienced in a wind tunnel, it is higher for a rotating machinery or wind turbine rotors (Hernandez et al., 2012). Transition analysis performed for wind tunnel experiments in controlled conditions includes measurements on wind turbine airfoils

equipped with pressure taps and sensors, balance system and a wake rake (Ceyhan et al., 2017); infrared thermography (Joseph et al., 2016); rotating turbine blade equipped with pressure sensors, strain gauges, balance system and particle image velocimetry (Schepers and Snel, 2007), rotating wind turbine and wind turbine blade experiments by oil visualization, stethoscope and flush-mounted unsteady pressure sensors (Lobo et al., 2018), wind turbine airfoil with pressure sensors and high frequency microphones (Özçakmak et al., 2019).

In addition to the DAN-AERO experimental campaign (Madsen et al., 2010) (Troldborg et al., 2013), of which the current analysis is based on, there have been other field experiments on boundary layer transition on rotating wind turbine blades using microphones glued on the surface (Van Ingen and Schepers, 2012), hot film and pressure tubes (Schwab et al., 2014) (Schaffarczyk et al., 2017), microphones on the suction side in addition to the ground based thermographic cameras (Reichstein et al., 2019). All these experiments pointed to a fact that more field experiments are needed on the wind turbine blades in

order to characterize the transition behaviour with inflow turbulence and rotational effects. Moreover, determining the relevant frequency ranges for the atmospheric turbulence and the occurrence of the T-S waves under real atmospheric conditions is needed in order to investigate the combined effects of the turbulent wind and the blade rotation on transition. Modern wind turbines usually operate in wind farms where the inflow is affected by the wake of the upstream turbines. They are also exposed to high free-stream atmospheric turbulence and wind shear. The differences in the transition behaviour of the same

airfoil section tested in a controlled 2-D wind tunnel environment and 3-D field experiments at real operational conditions is discussed in a previous work and it is seen that full rotor blade section exhibits different transition characteristics than in the 2-D case (Madsen et al., 2019). The difference between the design conditions for rotor and airfoils and the real operating conditions leads to inaccurate predictions of the loads and the performance.

In this study, the transition characteristics of the LM 38.8 blade on the NM80 2.3MW wind turbine is analysed by field

experiments (DAN-AERO project), and computations with the DTU in-house CFD EllipSys code (Sørensen, 1995) (Michelsen, 1992),(Michelsen, 1994). The present experimental analysis is based on high frequency microphone measurements that enables acquiring data at higher sampling frequencies and allows a higher resolution (with the number of microphones placed on both upper and lower surfaces) than the previous studies. This paper is focused on the analysis of the DAN-AERO 3-D transition rotor measurements in a wind farm and the validation of the transition models in the EllipSys3D CFD in-house solver using this

experimental data. The atmospheric turbulence, wind shear and wake effects on transition behaviour of a wind turbine blade



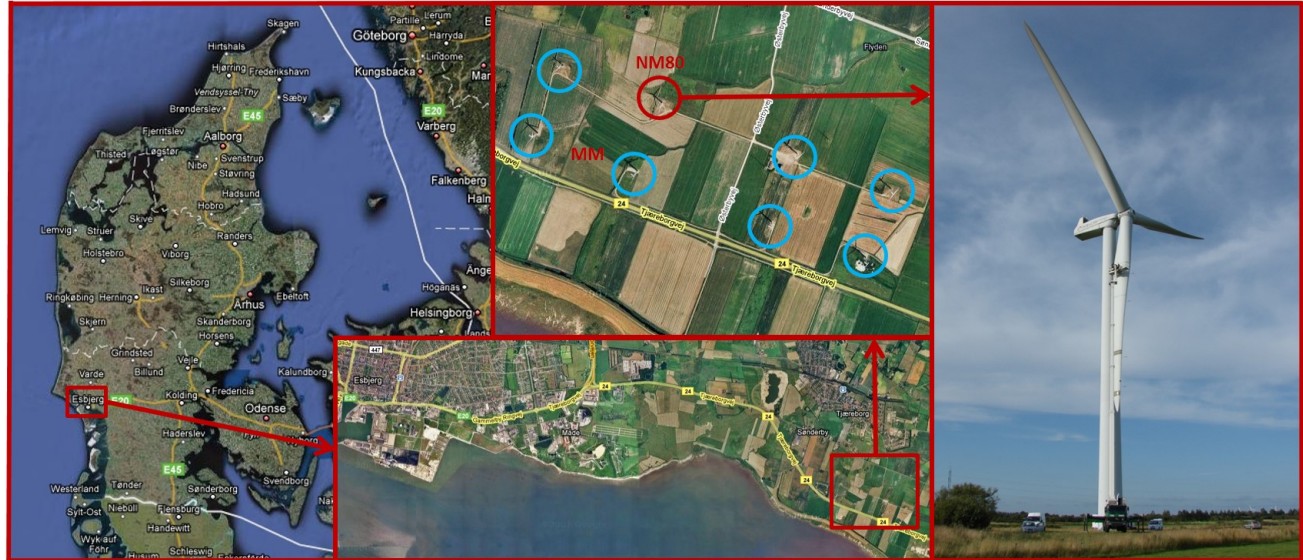

**Figure 1.** Map of the site approx. 10 km south-east of Esbjerg (© Google Maps 2007). The test turbine, NM80 is situated in a small wind farm at Tjæreborg along with 7 other wind turbines of size around 2 MW. The other turbines are circled with blue and the test turbine is marked by the red circle. The meteorology mast is shown by the 'MM' denotation in the picture.

section is discussed. The effect of these parameters on the effective angle of attack and velocity on the blade section as well as their potential direct effect on transition is discussed. Comparison of the field experiments with the CFD simulations enlightens the transition behaviour of the wind turbine blades and enables improvement of the design and aerodynamic prediction tools.

## 2 Field Experiments : Set-up and Instrumentation

The main objective of the DAN-AERO project was to establish an experimental database for aerodynamic, aeroelastic and aeroacoustic issues that are significant for the design and operation of MW size wind turbines (Bak et al., 2010). The laminar-turbulent transition investigation of this campaign contains both 2-D wind tunnel tests (Madsen et al., 2010) as later analysed by (Özçakmak et al., 2018) (Özçakmak et al., 2019), and 3-D field experiments (Troldborg et al., 2013).

In this study, field experiments are analysed in order to investigate the laminar-turbulent transition characteristics of a 3-D

rotor blade. The tested turbine is placed at a wind farm in Tjæreborg, Denmark, which consists of 8 turbines in 2 rows. The test turbine is a 2 MW NM-80 wind turbine with LM-38.8 blade. The rotor diameter is 80 meters. The site and the test turbine (denoted as 'NM80') is presented in Figure 1. The wake cases presented in this paper are from an upstream wind turbine that is located around 6 rotor diameters (6D) upstream of the test turbine.

The rotational speed, yaw, pitch and rotor azimuth angles are measured at the nacelle. In addition to the pressure taps placed

at four different sections on the blade, 56 high-frequency microphones are installed about 1 mm below the blade surface at a section 36.9 meters from the hub (3.1 m from the tip of the blade). The same section is also equipped with a pitot tube for





**Table 1.** Instrumentation and Data Acquisition Parameters

| Measurement device | Sampling rate | Acquisition time |
|---|---|---|
| Sensors on the turbine and MM | 35 Hz | 10 minutes |
| Pressure taps | 100 Hz | 9 minutes 30 seconds |
| High-freq. microphones | 50 kHz | 10 seconds |

measuring the relative velocity. The pressure taps are placed 36.8 m from the hub (3.2 m from the tip) next to the microphones. The wind direction and wind speed information of the inflow is obtained from the anemometers and wind vanes placed at the meteorological mast (denoted as 'MM') 2.5 diameter far from the test turbine, see Figure 1. At some specific wind directions, the test turbine is in the wake of the upstream turbine, effect of which is also discussed in this study. Both wake and no-wake

conditions are analyzed.

The angle of attack values of the field experiments presented in this article are derived from the normal force on the blade. The correlation between the normal force in the experiments and the angle of attack is generated by the HAWC2 (Horizontal Axis Wind turbine simulation Code 2nd generation)(Larsen and Hansen, 2007) simulations which is based on the principle blade element momentum theory with an aero-elastic model of NM80 turbine using existing polars.

The acquisition properties of the instruments on the test blade and the meteorological mast are listed at Table 1.

Different 10 minute series from 2 different days are used in this study with corresponding 10 second series of microphone acquisition. The turbine operated at two different pitch settings: $1.25°$ and $4.75°$ degrees. The pitch angle is defined as positive towards stall.

The blade section that is equipped with the high frequency microphones features a NACA 63-418 airfoil with a chord length

of 1.24 meters. The turbine is operated at a constant rotational speed of 16.1 revolutions per minute(rpm) and at a fixed pitch setting. The Reynolds number is around 5 million for the tested blade section. The region of the atmospheric boundary layer where the turbines operate has varying levels of free stream turbulence. This contributes to large external disturbances in the boundary layer of the blades. The degree of turbulence in the wind varies greatly on short time scales. Therefore, for the inflow velocities, 10 second datasets from MM measurements are used, which corresponds to microphone acquisition times.

The ABL velocity profiles are averaged over 10 minutes (a) and over 10 seconds (b) corresponding to the exact time instance of the microphone dataset, presented in Figure 2 with their standard deviations as shaded areas. In 10 second average profiles, for some of the selected cases, an increase in the velocity is observed at 57 meters from the ground where the turbine hub is approximately located.

The 2-D results shown in this paper originates from the wind tunnel experiments of the DAN-AERO project conducted in the

LM wind tunnel with a manufactured airfoil identical to the blade section on the wind turbine, details of which are explained in a previous study (Özçakmak et al., 2019).



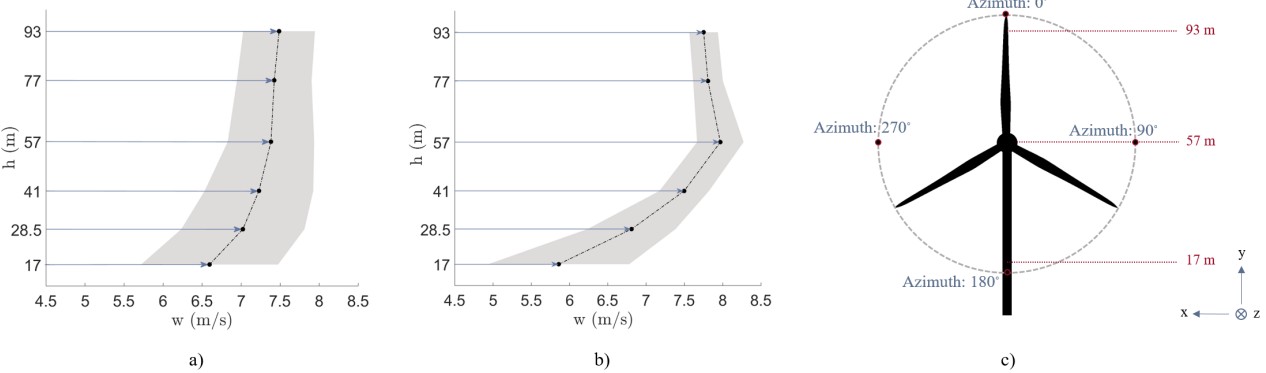

**Figure 2.** ABL velocity profiles: a) 10 minutes average, b) 10 seconds average corresponding to microphone acquisition time; c)Azimuthal placement and the heights of the blade cross-section at each azimuthal location

## 2.1 Data processing

The pressure fluctuations in time domain (10 second series), obtained from high frequency microphones placed chordwise on the blade section, are analyzed in the frequency domain by fast Fourier transform (FFT) analysis. The sampling frequency of the data is 50 kHz acquired over 10 seconds. The data is divided into smaller time segments of 0.0410 seconds. The window

size of 4096 is used with a 50 % overlap. For each time segment, the power spectral density (PSD) and the integrated PSD is calculated.

    The PSD of the pressure fluctuations obtained from the microphones are integrated in a frequency interval from f1=2 kHz to f2=7 kHz(see Equation 1). The integration within a certain frequency range gives the standard deviation ($\sigma$), which represents the total energy of the pressure fluctuations. A sudden chordwise increase in the pressure level ($L_p$) is considered as an

indication of the transition location. The reference pressure $P_{ref}$ is assigned to the value of 20 $\mu$ Pa. More details on this method and the selection procedure of the frequency interval for the integration can be found in a previous work by Özçakmak et al. (2019).

$$P_{rms}{}^2 = \int_{f1}^{f2} PSD\,df \quad , \qquad L_p = 20 \cdot log_{10}(\frac{P_{rms}}{P_{ref}}) \quad , \qquad \sigma = \sqrt{P_{rms}{}^2}. \tag{1}$$

    The transition location on the upper and lower surfaces is detected by the highest chordwise derivative of the pressure level

as in Equation 2. The derivatives that are above a threshold level of 250 dB are selected for transition detection.

$$x_{tr} = x \rightarrow \; max(\frac{dL_p}{dx}) \tag{2}$$





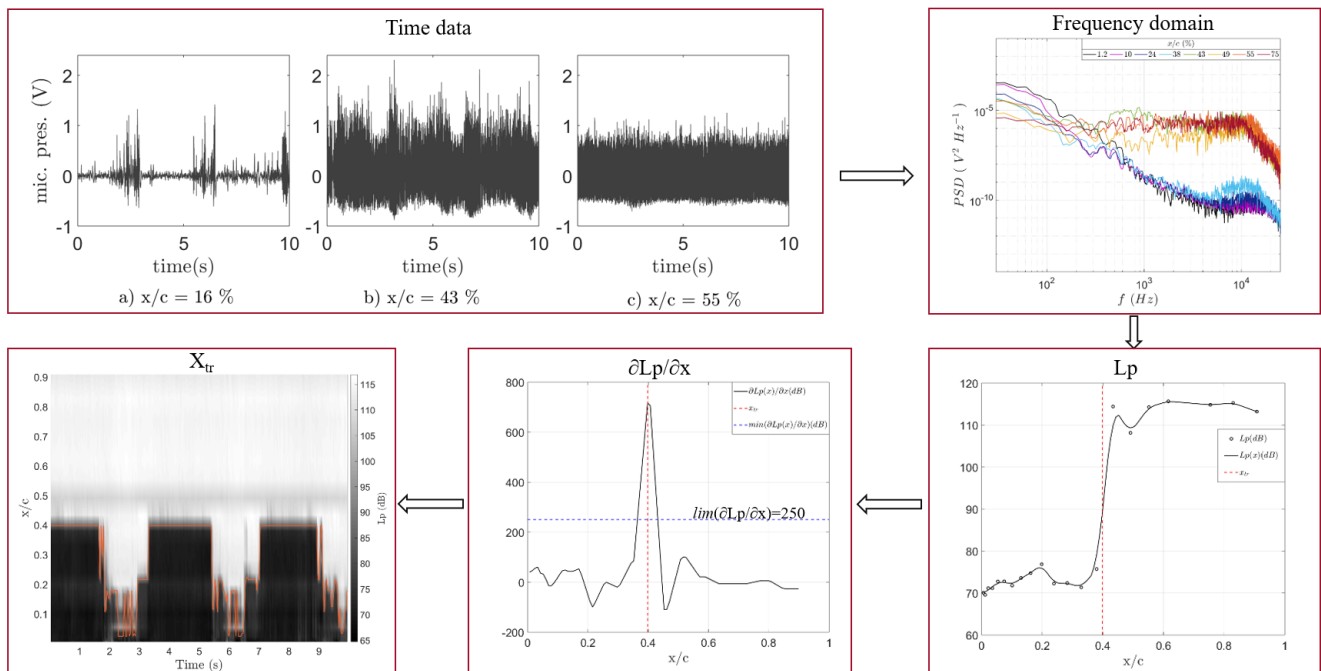

**Figure 3.** Transition detection methodology from the high frequency microphones

The transition detection method is illustrated in Figure 3. The laminar, transitional and the turbulent flow in time series and in frequency domain is illustrated at the top of the figure and the chordwise increase of the integrated PSD and the derivative of Lp is illustrated at the bottom. The spectrogram analyses are also performed by dividing the data length (L) into k columns. The final detected transition locations can be seen from the spectrogram at the bottom left figure.

## 3  CFD Computations

EllipSys3D is an in-house CFD solver for incompressible Navier-Stokes equations in general curvilinear coordinates by a multiblock finite volume discretization, here applied in RANS mode. Rhie-Chow (Rhie, 1982) interpolation is used in order to avoid odd/even pressure decoupling. The third order QUICK (Quadratic Upstream interpolation for Convective kinematics) upwind scheme is used for the convective terms. SIMPLE(Semi-Implicit Method for Pressure-Linked Equations) algorithm is

10 used to enforce the pressure/velocity coupling (Patankar and Spalding, 1983) (Patankar, 1980). The Message-Passing-Interface (MPI) is used to parallelize the code for executing on distributed memory machines with non-overlapping domain decomposition (Sørensen et al., 2011).

Mesh generation is done by the 3-D hyperbolic grid generation program HypGrid3D (Sørensen, 1998). The rotor geometry and the boundary layer of the blade surface is resolved by O-O mesh configuration. The grid consist of 256 cells in the





chordwise direction, 128 cells in the spanwise and 128 in the normal direction, with cells size ensuring y+ value less than 1. The mesh has around 14 million cells. The far field boundary is located around 10D away from the rotor in all directions. Due to the high tip speed ratio, the simulations are performed by transient computations with 1200 steps per revolution. At each time step, the momentum equations are solved and a pressure correction equation is used to satisfy the continuity constraint.

This process is repeated until a convergent solution is reached, and all the terms are evaluated at the next time level when the subiterations are finished.

The turbulence is modelled by the $k - \omega$ SST eddy viscosity model (Menter, 1993). The boundary layer transition prediction on the rotating wind turbine blade section is performed by the $e^N$ transition method based on linear stability theory.

In the CFD simulations, the $e^N$ transition method is applied for several flow conditions representing the field experiments.

The EllipSys3D rotor simulations are performed at several free stream velocities representing the ABL conditions at 3 different heights, considering both the mean and the standard deviations during the microphone acquisition time. Both the $e^N$ natural transition model with different amplification factors (N) and the bypass transition model at the corresponding atmospheric turbulence intensity values from the field experiments are simulated.

The angle of attack from the EllipSys3D simulations on the blade section is determined from the annular averaging method

by determining the local induced velocities at the blade (Hansen et al., 1997) (Johansen and Sørensen, 2004). In order to calculate the induced axial velocity, $V_{ind}$, at the rotor plane, annular averaging of the axial velocity at several upstream and downstream locations at a given radial location (in this case, the same blade section as in the microphone experiments) is performed. Then, $V_{ind}$ is found by the interpolation of these averaged streamwise velocities to the rotor plane. Lagrangian polynomial interpolation is used to determine the velocity at the rotor location ($z = z_0$) as in Equation 3.

$$f(z_0) = \sum_{i=1}^{N} \left[ f(z_i) \left( \prod_{j=1, j \neq i}^{N} \frac{z_0 - z_j}{z_i - z_j} \right) \right] \tag{3}$$

Having calculated $V_{ind}$, the effective local flow angle ($\alpha$) is found from Equation 4, where $\omega$ is the angular velocity, $R$ is the distance to the hub from the measured tested section on the blade, $\theta$ is the combined pitch twist angle.

$$\alpha = tan^{-1} \left( \frac{V_{ind}}{\omega \cdot R} \right) - \theta \tag{4}$$

The Fieldview (2017) software is used in order to postprocess the EllipSys3D simulation results, to extract the annular

averages of the axial velocity and for the flow visualization.

### 3.1  EllipSys3D semi-empirical $e^N$ transition model

Transition to turbulence in EllipSys3D is governed by the semi-empirical $e^N$ model (Drela and Giles, 1987), which is based on linear stability theory. The conventional $e^N$ method is a semi-empirical method not considering receptivity. In the semi-empirical method, while linear stability analysis is done for the governing equations, the transition is assumed to take place

when N reaches a previously correlated value from the experiments. Therefore, the empirical part comes from the N value at



transition, which makes the model semi-empirical. In the $e^N$ method , the amplification of small disturbances are calculated for several frequencies, and the spectrum of the most amplified ones is identified. The critical N factor for each type of flow is determined empirically, and the transition point is detected from this empirical value of the critical N factor.

In the EllipSys3D code, the boundary layer parameters (the displacement thickness $\delta^*$, momentum thickness $\theta$ and the shape factor $H$ in Equation 5) should be known to determine the occurrence of transition. $U_e$ is the velocity at the outer edge of the boundary layer which is determined from the Navier-Stokes computation, and $y$ is the direction perpendicular to the wall/surface. The boundary layer thickness $\delta$ is defined as the location where $u = 0.99U_e$.

$$\delta^* = \int_0^\infty (1 - \frac{u}{U_e})dy \quad , \quad \theta = \int_0^\infty \frac{u}{U_e}(1 - \frac{u}{U_e})dy \quad , \quad H = \frac{\delta^*}{\theta} \tag{5}$$

Although, these boundary layer parameters can be calculated by integrating the velocity profile from the Navier-Stokes equations, it is shown by Stock and Haase (1999) that it requires an excessively fine computational grid. Therefore, the boundary layer parameters are found from the von Karman boundary layer equations (Von Kármán, 1921).

The momentum integral equation (Equation 6) and the combination of the kinetic energy equation with von Karman's momentum equation (Equation 7) are solved for $H$ and $\theta$. $U_e$ is determined from the Navier-Stokes computation. In each iteration, these equations are started from the stagnation line and integrated downstream on the surface until the transition point is found.

$$\frac{Cf}{2} = \frac{1}{U_e^2}\frac{\partial}{\partial t}(U_e \delta^*) + \frac{\partial \theta}{\partial x} + (2 + H)\frac{\theta}{U_e}\frac{\partial U_e}{\partial x} + \frac{\theta}{U_e}\frac{\partial U_n}{\partial n} \tag{6}$$

$$2C_D = \theta \frac{\partial H^*}{\partial x} + H^*\left((1 - H)\frac{\theta}{U_e}\frac{\partial U_e}{\partial x} + \frac{Cf}{2}\right) \tag{7}$$

$C_D$ is the kinetic energy dissipation coefficient ($C_D = \frac{D}{\rho U_e^3}$) and $D$ is the dissipation per unit area. The $U_n$ term, the velocity normal to the wall, comes from the assumption of axial symmetry. These equations are presented in terms of boundary layer parameters and the skin friction coefficient $C_f$, where $x$ is the horizontal direction, $\tau_w$ is the wall shear stress, $\rho$ is density, $\mu$ is the dynamic viscosity, $\theta^*$ is the kinetic energy thickness, and $H^*$ is the energy thickness ratio as shown in Equation 8.

$$\tau_w = \mu\frac{\partial u}{\partial y} \quad , \quad C_f = \frac{t_w}{\frac{1}{2}\rho U_e^2} \quad , \quad \theta^* = \int_0^\infty \frac{u}{U_e}(1 - \frac{u^2}{U_e^2})dy \quad , \quad H^* = \frac{\theta^*}{\theta} \tag{8}$$

Closures for $C_f$, $C_d$ and $\theta^*$ are calculated based on Falkner-Skan velocity profiles. The stagnation line is normally found as the location where the pressure coefficient based on relative velocity is equal to 1. In order to ensure that the entire stagnation line is found at each time step, Hiemenz flow across the stagnation line and Blasius flow along the stagnation line are assumed.



By this way, realistic initial values for $H$ and $\theta$ are obtained and the stagnation line can be located between two computational cells.

The development of the imposed wave perturbations' amplitude is computed along the boundary layer based on spatial analysis. A check is carried out during the integration of the boundary layer equations to determine if the disturbances are amplified or damped. As neutral stability is passed, amplifications are determined for a range of temporal frequencies.

The $N$ factor is the natural logarithm of the ratio of the disturbance amplitude at a specific location to its amplitude at the neutral stability point. Integration of the $N$ amplitude is carried out until it reaches a certain value for which transition is said to occur. This value is set according to the turbulence degree that is present in the experimental conditions for comparison with the simulations. In order to build a relation between the turbulence level and the amplification factor, Mack's expression (Mack, 1977) is used as an estimate.

Transition to turbulence is handled by the intermittency factor ($\gamma$) when solving the Navier Stokes equations. The eddy viscosity ($\mu_T$) obtained from the turbulence model is multiplied with the intermittency factor which controls the effective viscosity ($\mu_{eff}=\mu+\gamma\mu_T$), where $\mu$ is the molecular viscosity.

$$Intermittency \begin{cases} \gamma = 0, & \text{laminar flow} \\ 0 < \gamma < 1, & \text{gradually increases to 1 after transition} \\ \gamma = 1 & \text{turbulent flow} \end{cases} \tag{9}$$

The intermittency factor is calculated from Equation 10. This equation is obtained by combining the statistical theory for transitional flow by Emmons (1951), and the expression that represents the production rate with Gaussian distribution using the Dirac delta function by Dhawan and Narasimha (1958) and the Chen and Thyson (1971) formulation:

$$\gamma = 1 - exp\{-(x-x_{tr})^2 \left(\frac{U_{e,Tr}}{\nu}\right)^2 \widehat{n}\sigma\} \tag{10}$$

where the subscript $tr$ is the transition onset, $\nu$ is the kinematic viscosity, $\sigma$ is here the spot propagation rate, and $\widehat{n}$ is the non-dimensional spot formation rate, $\widehat{n} = n \cdot \nu^2/U_{e,Tr}{}^3$ (Mayle, 1998).

The intermittency factor is calculated on the surface and then solved for the entire boundary layer and wake within the transport equation (Michelsen, 2002).

$$\frac{\partial\gamma}{\partial t} + \frac{\partial U\gamma}{\partial x} + \frac{\partial V\gamma}{\partial y} + \frac{\partial W\gamma}{\partial z} = S \tag{11}$$

where the source term, $S$, is obtained by evaluating the transport terms for previously determined intermittency values.

### 3.1.1 Bypass transition model

When the amplitude of the disturbances are strong, such as for high free stream turbulence or large roughness elements, the the linear stages of the transition process is bypassed. In this case, transition happens in the absence of T-S waves and the disturbances are amplified by a non-linear process.





The approaches for modelling bypass transition in industry involves low Reynolds number turbulence models, and models using experimental correlations that relates free-stream turbulence intensity to transition Reynolds number based on momentum thickness $Re_{\theta t}$ (Reza and Amir, 2009).

The $e^N$ method accurately predicts the transition for free stream turbulence levels , 0-2 % for T-S dominated transition, but
for higher levels it is bypassed (Biau et al., 2007). The $e^N$ method in EllipSys3D can be used together with a bypass criteria. For the bypass transition model, Suzen and Huang (2000) empirical model is used.

Abu-Ghannam and Shaw (1980) suggested that for the attached flows, transition onset can be obtained by correlating $Re_\theta$ to the free stream turbulence intensity. By maintaining the strong features of this correlation in adverse pressure gradient regions, more sensitive response to the favourable pressure gradients is obtained by Suzen et al. (2002) by re-correlating the transition
criterion to the free stream turbulence intensity and acceleration parameter $K_t$.

$$Re_{\theta_{tr}} = (120 + 150Tu^{(-2/3)})coth\left[4(0.3 - 10^5 K_t)\right] \tag{12}$$

Here, $K_t$ is the minimum value of the acceleration parameter in the downstream direction (Michelsen, 2002), which can be expressed as $\nu/U_t{}^2 (dU/dx)_t$ where $U_t$ is the boundary layer velocity at onset of transition (Suzen et al., 2002). $Tu$ is the turbulence intensity at the transition onset. Under high turbulence intensity conditions, this correlation fits well with adverse
pressure gradient regions. In EllipSys3D, for the bypass transition cases, where turbulence levels are high, this correlation is used and the criteria for natural and bypass transitions are checked simultaneously in the code. The higher of the two intermittency factor is used. Moreover, separation induced transition is also checked with a bubble model inside the boundary layer solver.

### 3.2 Numerical Set-Up

The computations are performed for three different grid levels for the 3-D simulations, and five different grid levels for the 2-D case. The grid independence is ensured and the results of the finest grid are presented.

The 3-D full rotor simulations are performed as transient calculations with 1200 steps per revolution for all grid levels. The problem is approximately axisymmetric. The CFD Simulation input parameters are listed in Table 1 for the finest grid level. The input $k$ is the turbulent kinetic energy and $\omega$ is the specific dissipation for the turbulence model. Several free-stream
velocities are simulated from 5.5 m/s to 8.5 m/s. It should be noted that, in the experiments, the wind speed and the turbulence levels vary as a function of the blade azimuth due to the wake and wind shear. Therefore, these local inflow conditions are represented by individual CFD simulations with various wind speeds and T.I. since it is difficult to simulate spatially varying inflow conditions only by a single simulation. Moreover, at each simulation, different amplification factors ($N = 0.15, 3$ and $7$) are used in the natural transition model to represent different inflow scenarios.
The turbulence is quantified by the turbulence intensity (T.I), which is the standard deviation of the relative velocity divided by the average relative velocity over 10 minutes in this case. Ten minutes average of the velocity data obtained from the pitot tube on the blade is used in order to obtain the T.I. values. Various T.I. values of 2.8, 3.8 and 6.8 % are used as an input to the





**Table 2.** CFD Simulation parameters

| Parameter | Value |
|---|---|
| Density | $1.23 \quad kg/m^3$ |
| Viscosity | $1.83 \cdot 10^{-5} \quad kg/ms$ |
| $k$ T.K.E. (turbulence model) | $0.1 \quad m^2/s^2$ |
| $\omega$ (turbulence model) | $1 \cdot 10^6 \quad 1/s$ |
| Rotational speed | $1.7 \quad rad/s$ |

bypass transition model in the CFD computations. The transition point is selected to be the first location where $\gamma \geqslant 0.025$ for both of the transition models.

## 4 Results and Discussion

### 4.1 Atmospheric effects on transition

The flow on the blade is affected by many parameters in the free atmosphere compared to an airfoil in a steady flow in a wind tunnel. The atmospheric flow has a substantial effect on the performance and loading of the turbines. The shear in the atmospheric boundary layer and the inflow turbulence in combination with the rotational effects, creates a significant deviation of the aerodynamic characteristics on the blade compared to the wind tunnel flow conditions. The effective angle of attack and velocity varies as a function of azimuth position with these parameters and causes the transition position to move constantly

during rotation. These multiple effects on transition are not easy to analyse separately. For instance, the large scales of the inflow turbulence affect the effective angle of attack on the blade section, but may also have a direct effect on the transition location as in the wind tunnels. Thus, analysis from the experiments in this study aims to analyse these individual effects on the transition behaviour.

The spectrogram of the chordwise pressure levels on the airfoil profile tested in the wind tunnel (Figure 4-a) and wind turbine

blade section (Figure 4-b) from the field experiments, featuring the same airfoil profile are presented. The transition regions can be seen by the sudden chordwise increase of the pressure levels. Due to the low inflow turbulence in the wind tunnel (T.I.=0.1%), no time variations of the transition position is observed for the airfoil profile. On the other hand, the transition location changes significantly for the rotor blade section through one revolution for the high pitch case (p=4.75°) (Figure 4-b) , deviating from the 2D case (Figure 4-a).

In order to analyze the causes of this variation, different inflow turbulence levels are investigated. As the PSD is integrated from 2 kHz to 7 kHz for transition detection (presented in Figure 5-b), several frequency intervals of integration are attempted to identify the inflow turbulence from the microphone signals. It is found out that the microphones closer to the leading edge have high energy content in the low frequency region. Therefore, the PSD is integrated in the frequency interval from 100 Hz

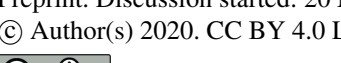





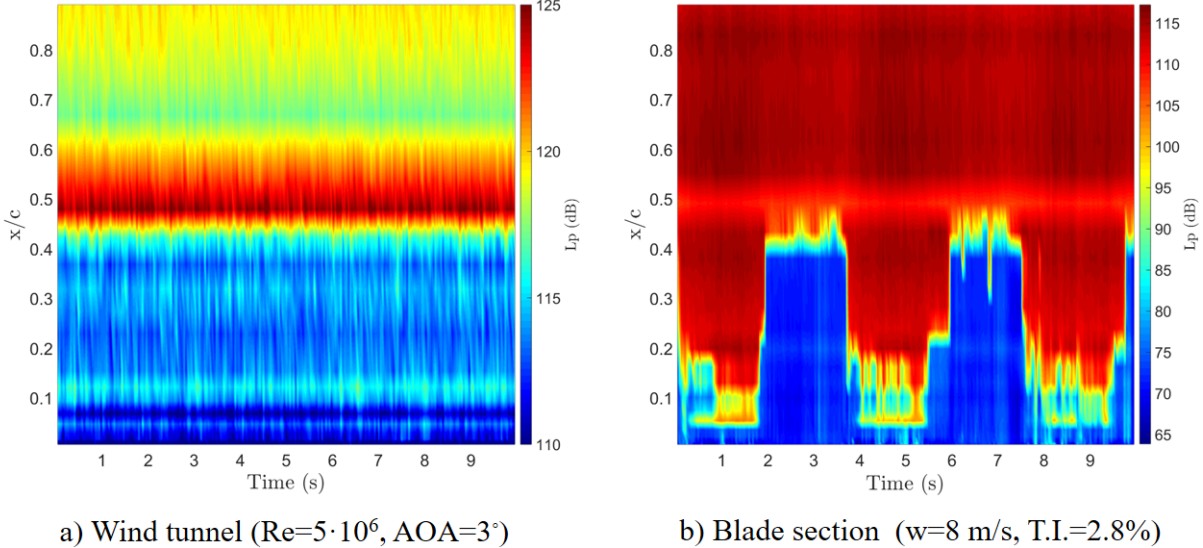

a) Wind tunnel (Re=5·10⁶, AOA=3°)    b) Blade section  (w=8 m/s, T.I.=2.8%)

**Figure 4.** Chordwise pressure level (Lp) spectrogram along the pressure side for airfoil tested at the wind tunnel(a) and for the blade section from the field experiments (b).

to 300 Hz. It can be seen from Figure 5-a that the leading edge microphones has high pressure levels on this frequency range capturing the pressure response to the inflow turbulence. For the quantitative comparisons, a microphone located very close to the leading edge (at $x/c = 2.2\%$) on the pressure side, in the laminar boundary layer, is selected to represent the inflow turbulence that the blade section is exposed to.

The inflow turbulence levels from the microphone analysis, the relative velocity obtained from the pitot tube on the blade section, and the angle of attack, which is derived from the forces, are presented together with the detected transition points as a function of the azimuthal angle in Figure 6. Two different cases from the measurements are shown in Figure 6. Figure 6 - left shows a partial wake case (9% wake-affected rotor area) with a free stream velocity ($U_\infty$) of 7.2 m/s. Figure 6 - right demonstrates a half wake case (48% of the rotor area covered with wake affected inflow) with $U_\infty$=8 m/s. Both cases have the

same pitch setting and the same T.I= 2.8% obtained from the meteorological mast. The wake shadow from the rotor view and the top view is also shown at the bottom of the figure for each case. The swept area that is influenced by the wake according to the wind direction was calculated by estimating the wake expansion depending on the velocity induction factor. Each line at $V_{rel}$, $AOA$, $Lp$ and $Xtr$ plots corresponds to a single revolution. For the 9% wake case, each revolution tends to have a similar behaviour, on the contrary, for the 48% wake case, there are some discrepancies observed between each rotation. For

both cases, it is noticeable that the decrease in relative velocity and angle of attack and the increase of the inflow turbulence leads the transition point to move closer to the leading edge at the pressure side. The tilt, yaw and wind shear effects on AOA and Vrel are analyzed by HAWC2 simulations. Yaw misalignment (which is the difference between the angle measured on the

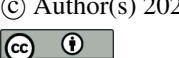



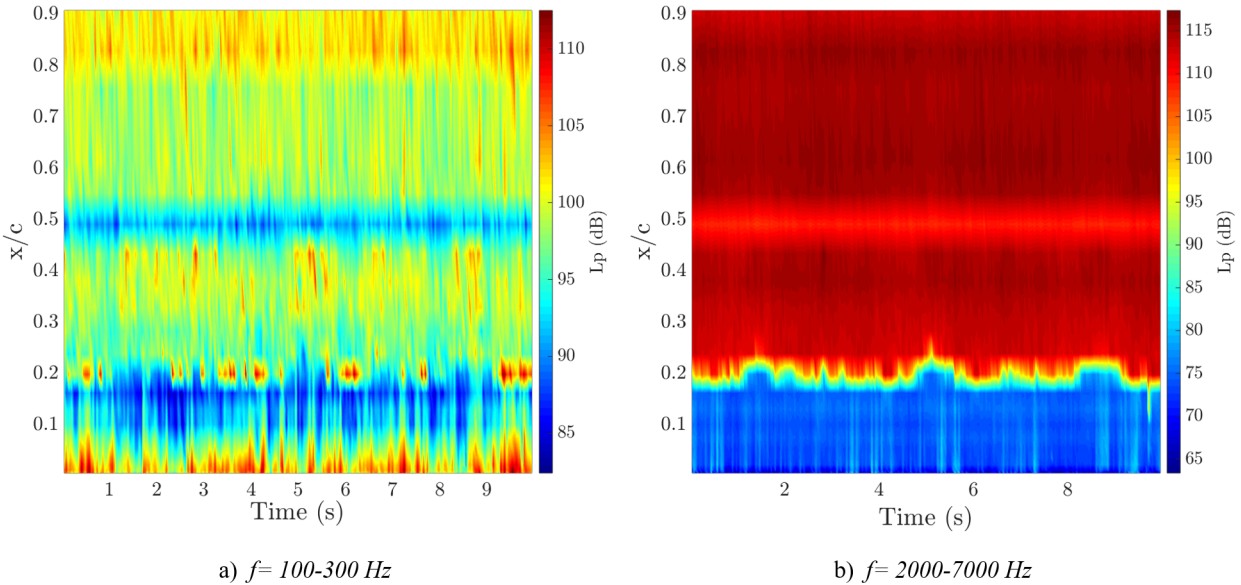

a) *f= 100-300 Hz*  b) *f= 2000-7000 Hz*

**Figure 5.** Spectrogram of the chordwise pressure levels obtained from frequency integration limits of a)100-300 Hz and b) 2-7 kHz on the pressure side, for the low pitch case (p=1.25°), w= 6.3 m/s , T.I.= 3.8%

nacelle and the wind direction measured from the meteorological mast at the same height) was checked for the cases presented in this paper. The mean absolute yaw error is found to be less than 5 degrees, and the effect of the maximum yaw on $V_{rel}$ and AOA change is found to be no more than 1% by HAWC2 analysis. Moreover, the wind turbine has a 5° tilt angle that causes 1% change in $V_{rel}$ and 0.2% change in AOA according to this analysis. Considering that the cases presented here are not under

5 a strong shear, and comparing those variations with the experiments, it can be concluded that the azimuthal behaviour of the relative velocity and the angle of attack is governed by the inflow turbulence, mainly from the wake of an upstream turbine.

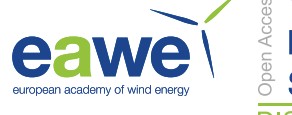
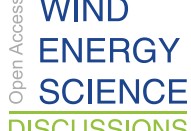

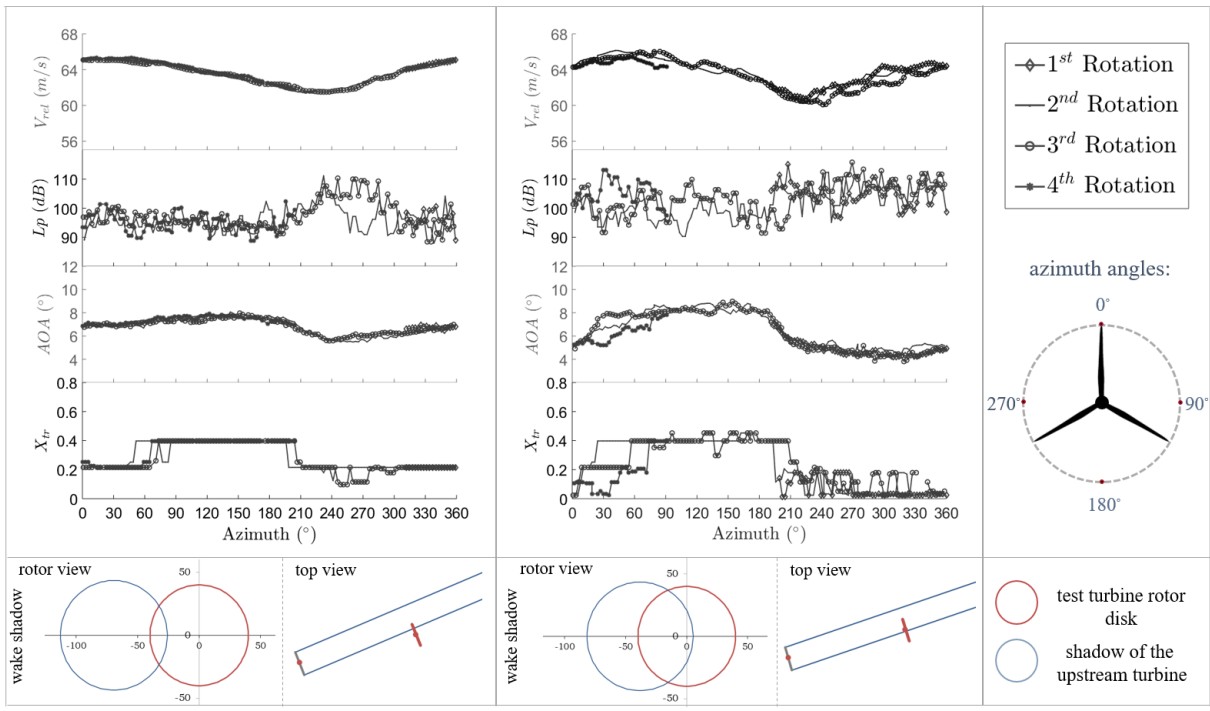

**Figure 6.** The relative velocity (Vrel), pressure levels (Lp) of the inflow turbulence, AOA, and detected transition points (Xtr) for the pressure side: (left) 9% wake shadow from an upstream turbine with $U_\infty$=7.2 m/s; (right) 48% wake shadow with $U_\infty$=8 m/s; (bottom) rotor view and top view of the upstream turbine and the test turbine.

In order to distinguish the effects of the AOA and the T.I., the data is divided into AOA bins. A previous study (Madsen et al., 2019) with data from the DAN-AERO experiments shows that for several angle of attack bins, there is a correlation between inflow turbulence and the transition location. Increasing the turbulence content in the range of 100-300 Hz moves the transition process closer to the leading edge at the pressure side. At the suction side, transition points are detected within the first 13% of

5 the chord, and no correlation could be established with the inflow turbulence.

### 4.2 Angle of attack effect on transition

The relative velocity, turbulence levels, AOA and the detected transition points on the pressure side for two different pitch cases are compared in Figure 7 as a function of time. These two cases belong to different measurement sets; a low pitch case (right) with 7 m/s free stream velocity with 2.6% T.I. and a high pitch case (left) with 7.2 m/s free stream velocity with 2.8% T.I. The

10 high pitch case is under 8.9% wake from the upstream turbine while the case with low pitch angle is under no wake conditions. Therefore, despite some slight differences, these two cases are approximately under the same inflow conditions, which allows





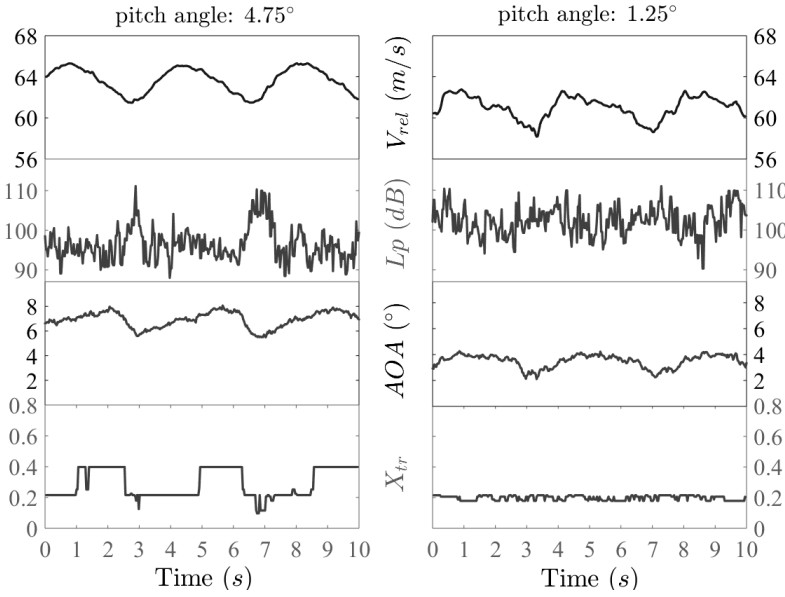

**Figure 7.** The relative velocity (Vrel), pressure levels (Lp) of the inflow turbulence, AOA, and detected transition points (Xtr) for the pressure side: (left) pitch=4.75°, 9% wake shadow from an upstream turbine with $U_\infty$=7.2 m/s; (right) pitch=1.25°, no wake case with $U_\infty$=7 m/s

analysing the effect of the angle of attack on transition. It can be seen that the transition location on the pressure side has a higher response to the rotational changes for the high pitch case. In the low pitch case, the transition point on the pressure side is closer to the leading edge than in the high pitch angle case. For the high pitch case, the regions where the transition points move closer to the leading edge around 20% of the chord corresponds to the regions of lower angle of attack values, as seen in
the low pitch case.

### 4.3 CFD Transition Locations

The vorticity structures from the rotor coloured by the axial velocity (w) (top left); the intermittency factor ($\gamma$) around the blade section, 36.8 meters from the hub, (top right) and on the blade for the pressure and the suction sides (bottom) are presented in Figure 8. It can be seen from the vorticity iso-surfaces that the problem is approximately axisymmetric. From the intermittency
factor visualization, the transition location on the upper and lower surfaces can be seen on the blade section (top right). The blue parts where $\gamma$ equals to zero are showing the laminar parts in the flow and the red regions( $\gamma$=1) show the turbulent parts. The $\gamma$ on the suction side (bottom) of the full blade shows that the transition point moves closer to the leading edge while going from root to tip, except the inner most part of the root. For the pressure side, less change in the transition location is observed as going from root to tip on the blade compared to the suction side. The case presented here is the blade with a pitch setting of
4.75°, a free stream velocity of 7.2 m/s and an amplification factor (N) of 3.



eawe
european academy of wind energy

WIND
ENERGY
SCIENCE
DISCUSSIONS

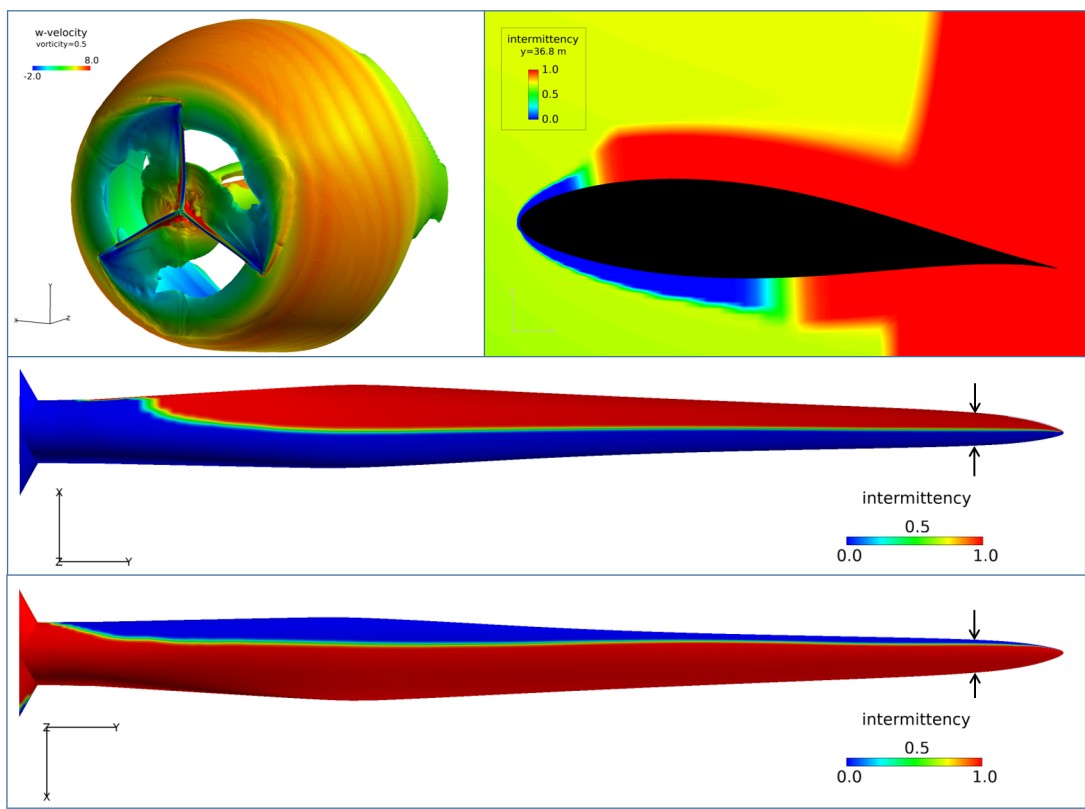

**Figure 8.** The vorticity iso-surfaces coloured by the axial velocity (top-left); intermittency($\gamma$) at the blade section at y=36.8 m from the hub(top-right); The $\gamma$ values on the pressure (middle) and suction sides (bottom) of the blade for $U_\infty$=7.2 m/s, N=3, p=4.75°. (Note that the free stream flow is in +z direction and the rotational speed in -x direction for the blade at azimuthal angle of 0°). (The blade section that is analysed in the current study is highlighted by the arrows on the blade.).

### 4.4 Comparison between CFD and Experimental Results

Since the effective angle of attack in the experiments is derived from the force measurements, in order to have a direct comparison, the experimental forces are compared with the forces obtained from the EllipSys3D simulations. The $Fx$ and $Fz$ forces from the simulations in global coordinates are transformed to local blade section coordinates in order to obtain the chord normal force on the blade as in the experiments. The 10 seconds data are extracted from the 10 minute measurements that correspond to the exact microphone acquisition time to represent the cross sectional force for the same case. This case is under $48\%$ wake shadow with $U_\infty$=8 m/s. The comparison is presented in Figure 9. Several CFD simulations obtained with various N factors and inflow velocities for natural and bypass transition models are combined to capture actual azimuthal variation behaviour of the experiments. The wake shadow falls in the azimuthal range from 200 to 340 degrees for this case from the measurements. While large scale vortices contained in the wake of the upstream turbine might mean higher turbulence intensities, there is a



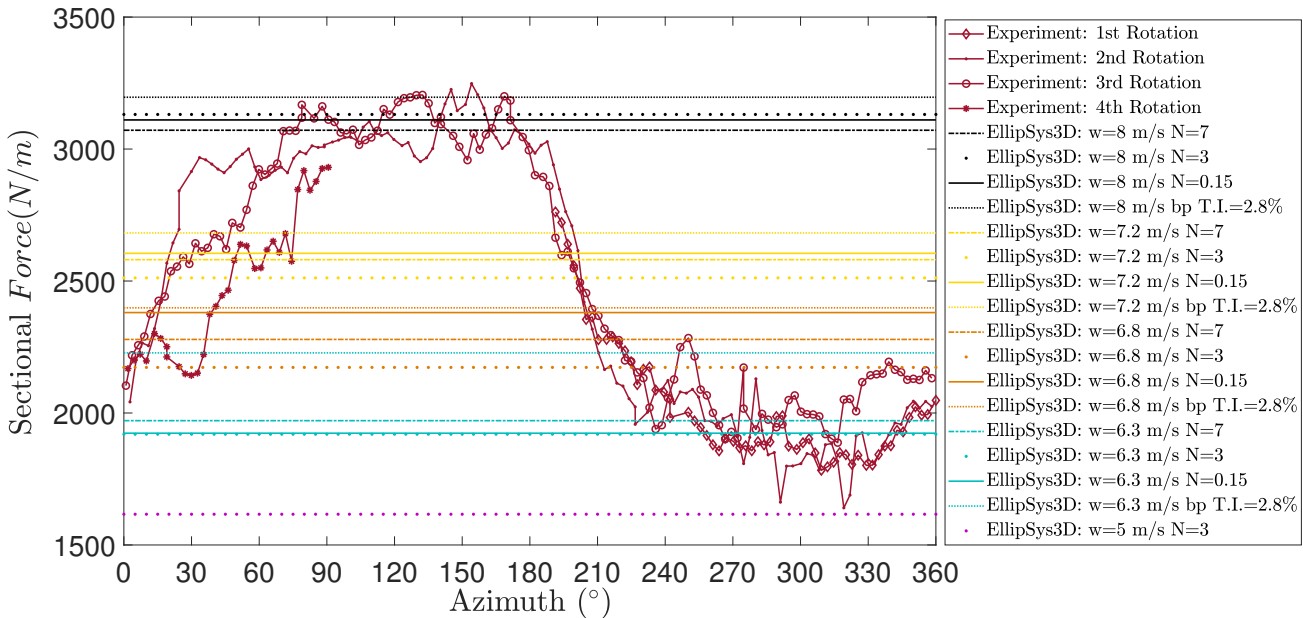

**Figure 9.** Sectional normal force versus azimuth angle from the DAN-AERO Experiments compared with EllipSys3D simulations with several inflow cases. (The CFD results are labelled by colors for the different freestream velocities (w); i.e. black: w=8 m/s, yellow: w=7.2 m/s, orange : w=6.8 m/s, turquoise: w=6.3 m/s, purple: w=5 m/s)

also a velocity reduction due to the energy extracted from the wind by the upstream turbine. Therefore, in the wake region, the experimental force shows agreement with the forces obtained from the simulations with a lower inflow velocity. It is also observed from the data that high wake cases introduces bigger amount of variation in the sectional force compared to low wake cases.

5    It is seen that the experimental force variation for 4 revolutions is comparable with the numerical results. For further valida-tion, the transition points are also compared for the pressure and the suction sides in Figures 10 and 11 respectively. Figure 9, 10 and 11 show the results for the same measurement dataset. Three different turbulence intensity levels are shown in most of the cases to cover several ranges of turbulence levels in the atmosphere.

It is seen from Figure 10 that the azimuthal angles that correspond to the wake inflow conditions match with the bypass
10   transition results. The main flow conditions measured from the meteorological mast is $U_\infty$=8 m/s and from the pitot tube on the blade; T.I= 2.8%, which corresponds to the N=0.15 with Mack's estimation. The regions where the transition point is around 20% of the chord is due to the effect of the decreasing AOA by decreasing relative velocity in the wake region. Moreover, the regions where the transition point moves closer to the leading edge, approximately to 3%, are the indication of the direct effect of the inflow turbulence on the transition point in addition to the decreasing AOA. In those regions, the amplification of the



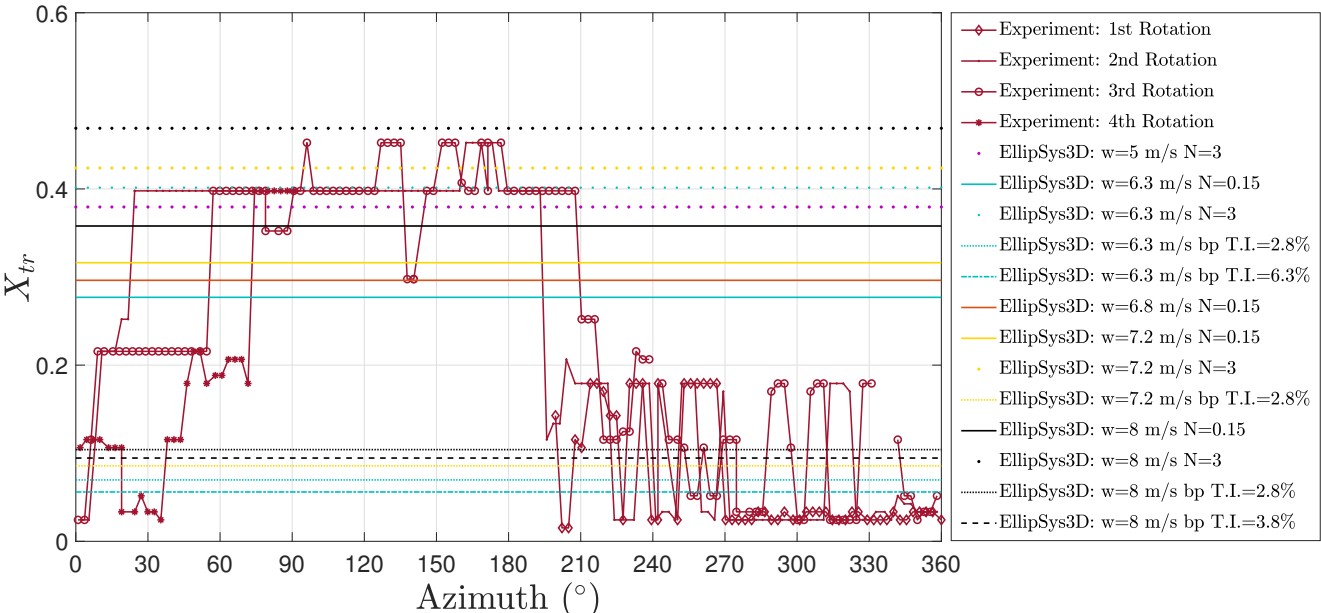

**Figure 10.** Experimentally detected transition points as function of azimuth angle and EllipSys3D simulations transition results for various different scenarios for the pressure side of the blade section. (The CFD results are labelled by colors for the different freestream velocities (w); i.e. black: w=8 m/s, yellow: w=7.2 m/s, orange : w=6.8 m/s, turquoise: w=6.3 m/s, purple: w=5 m/s)

small disturbances are bypassed. Considering the standard deviation of the measurements, EllipSys3D transition results covers most of the scenarios that the turbine is exposed to during rotation.

On the suction side, in Figure 11, the opposite behaviour of the transition point is noticeable with the angle variation. The regions where the transition point is closer to the leading edge corresponds to the high angle regions. The regions under wake
at higher azimuthal angles correspond to the decreasing $V_{rel}$ and AOA (see Figure 6-right) which moved the transition point further downstream. In this region on the suction side, although the turbulence intensity is increasing, the individual effect of the increasing AOA is more prominent than the effect of the inflow turbulence itself. However, variations between each rotation is also noticeable, which might be due to the inflow turbulence. It should be noted that for the suction side of the rotor blade section operating in real atmospheric conditions, although there are rotational changes in the transition point, $X_{tr}$, it is
considerably close to the leading edge, so the relative movement is not as prominent as in the pressure side and it is harder to reach a reliable conclusion.

The transition positions from the microphone measurements and the sectional forces derived from the pressure measurements are coupled. $Xtr$ on the pressure side as a function of the sectional normal force is presented in Figure 12. In the same way, the EllipSys3D transition point results are presented as a function of the sectional force obtained from the simulations. Each
15 EllipSys3D transition location-normal force point corresponds to a different simulation set-up where the input velocity and the

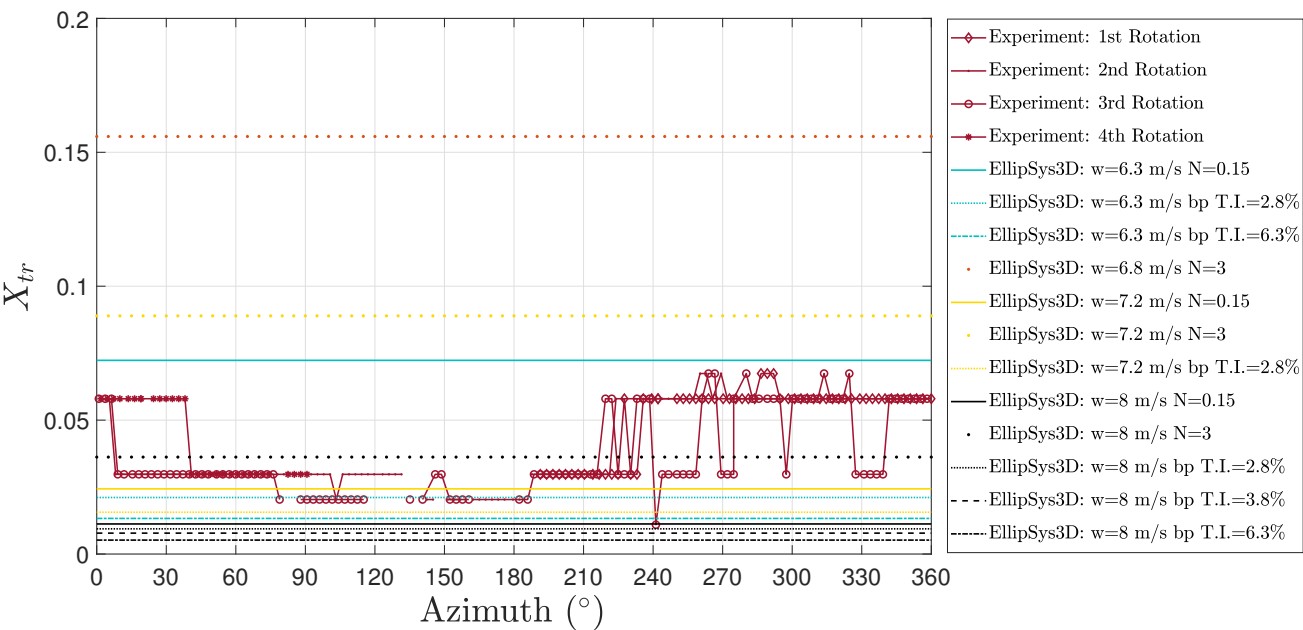

**Figure 11.** Experimentally detected transition points as function of azimuth angle and EllipSys3D simulations transition results for various different scenarios for the suction side of the blade section. (The CFD results are labelled by colors for the different freestream velocities (w); i.e. black: w=8 m/s, yellow: w=7.2 m/s, orange : w=6.8 m/s, turquoise: w=6.3 m/s)

amplification factor for the natural transition model and the turbulence intensity for the bypass transition model is varied. This comparison indicates that the simulation results are in line with the experiments although a considerable scatter is seen.

The pressure coefficient results from the pressure taps and those from the simulations are also compared. In the experiments, 4 blade sections are equipped with pressure taps and the current analysis shows the most outward section (next to the

5    microphones) that has the highest velocity change with the azimuth compared to the other sections. The pressure coefficient is calculated as follows:

$$C_p = \frac{p - p_\infty}{\frac{1}{2}\rho(V_\infty^2 + (r\omega)^2)} \tag{13}$$

where $V_\infty$ is the free stream velocity measured on the meteorological mast, r is the radial position of the section where the microphones are located and $\omega$ is the rotational speed of the turbine. The pressure coefficient from the experiments is obtained

10   by azimuthal averaging of the pressure measurements from a 570 seconds time data. For comparison, Cp values obtained by averaging through the full rotation is also presented in Figure 13. Moreover, both EllipSys2D and EllipSys3D results are presented and DAN-AERO experiments from the wind tunnel measurements (DAN-AERO 2D) are also included.

The Cp value from field experiments for $0°$ azimuthal angle and for the full rotation is presented in Figure 13-a. The EllipSys3D result, simulated with a free stream velocity of 7.2 m/s, fits well with the 3D experimental results. At this azimuthal




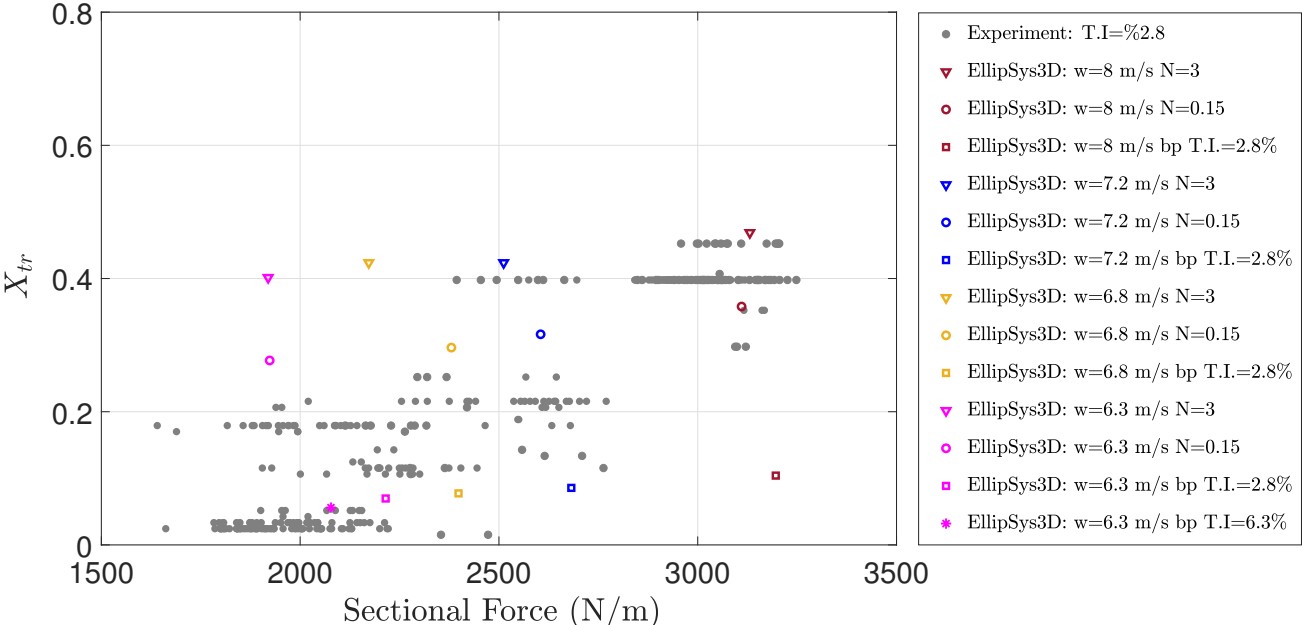

**Figure 12.** Experimentally detected transition points on the pressure side versus sectional normal force with various free stream velocity bins compared with EllipSys3D simulation results. (The experimental data is color coded according to the free stream velocity, see the legend, right of the Figure.)

position, the AOA values seen in the field experiments varies from 4 to 7.5 degrees. The 2D results presented here are for AOA= $4°$ and $5°$, at Reynolds number of 5 million. Furthermore, the 2D results for higher angle of attack values are also found to be still within the standard deviation of the pressure coefficient from the field experiments. Although there are some bumps on the Cp curve for the 2D simulations and both for the 2D and 3D experiments on the suction side due to the manufactured geometry, there are no visible bumps in the 3D simulations since the blades were generated with theoretical airfoil sections for the 3D simulations.

The Cp values obtained from 3-D simulations and experimental Cp value at $90°$ azimuthal angle is presented in Figure 13-b. EllipSys3D simulations for various free stream velocities observed during the acquisition time of the pressure measurements fit well with the 3D experimental results. In Figure 13-c, EllipSys3D results for fully turbulent, natural and bypass transition are shown for $270°$ azimuthal position. This is the region where there is wake affected inflow in this measurement set, therefore numerical results obtained for lower freestream velocities show agreement with the 3D experimental results.

In order to have a wider discussion, all the selected measurements from different days in the field experiments with different pitch settings, inflow velocities and T.I. under both wake and non-wake conditions are gathered and plotted as a function of the AOA. As explained in Section 3, for the EllipSys3D results, the effective angle of attack on the blade section is determined by annular averaging of the axial velocity method in order to compare the results with 2D simulations and experiments. Moreover, the AOA values derived from the force measurements in the 3D experiments with the detected transition points from the



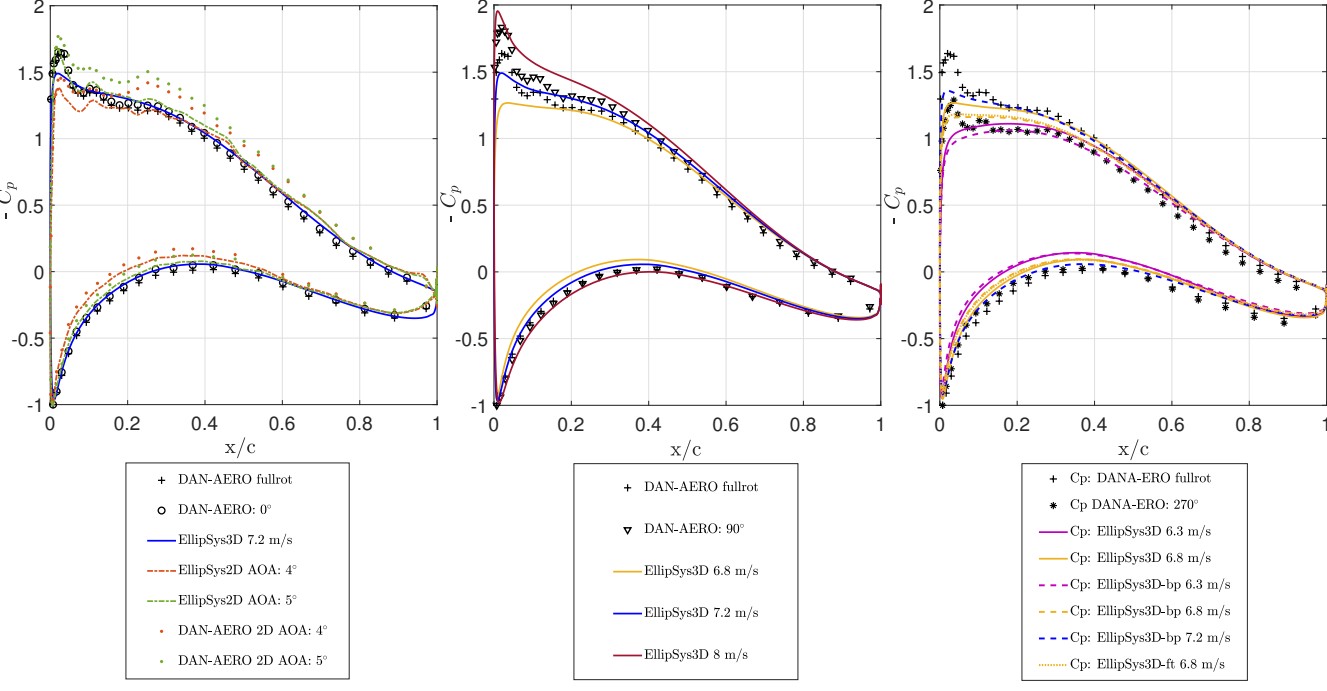

a) Azimuth: $0°$     b) Azimuth: $90°$     c) Azimuth: $270°$

**Figure 13.** Pressure coefficient (Cp) comparison of the 2-D and 3-D simulation and the experimental results for azimuth angle of $0°$ (a), comparison of the 3-D experimental results at $90°$ azimuth angle with numerical results featuring several free stream velocities (b), and comparison of 3-D experimental results at $270°$ azimuth angle with 3-D numerical results for fully turbulent ('ft'), natural and bypass transition ('bp') models (c). For each plot, Cp value obtained by averaging through full rotation is also presented and denoted as 'fullrot'.

microphone measurements are added to this comparison. The field experiment results contain measurements from 2 different days with two pitch (p) settings of $1.25°$ and $4.75°$ and both wake and non wake conditions. This data is binned according to the pitch setting and wake shadow ranges. Transition points as a function of the AOA for both 2D, 3D experiments and simulations are shown in Figure 14. The airfoil tested in the wind tunnel is manufactured identical to the rotor blade section surface geometry. Therefore, possible surface irregularities are also transferred. The reason of the sudden change of the transition point from AOA= $-6°$ to $-5°$ on the suction side in the 2D experiments instead of following a gradual pattern as in the EllipSys2D simulations might be due to these irregularities.

It can be seen from Figure 14-left (pressure side) that for the high AOA and high inflow turbulent cases, the transition locations are scattered around a larger percentage of the chord (from x/c=0.016 to x/c=0.46, approx. 44 % area) which creates a significant effect on the performance of the wind turbine. However, for lower AOA values under no wake conditions, the transition point does not move within more more $10\%$ along the chord during a revolution. The 2D simulations with N=7 and the wind tunnel experiments show good agreement on the pressure side. It is seen that there is a significant difference on the

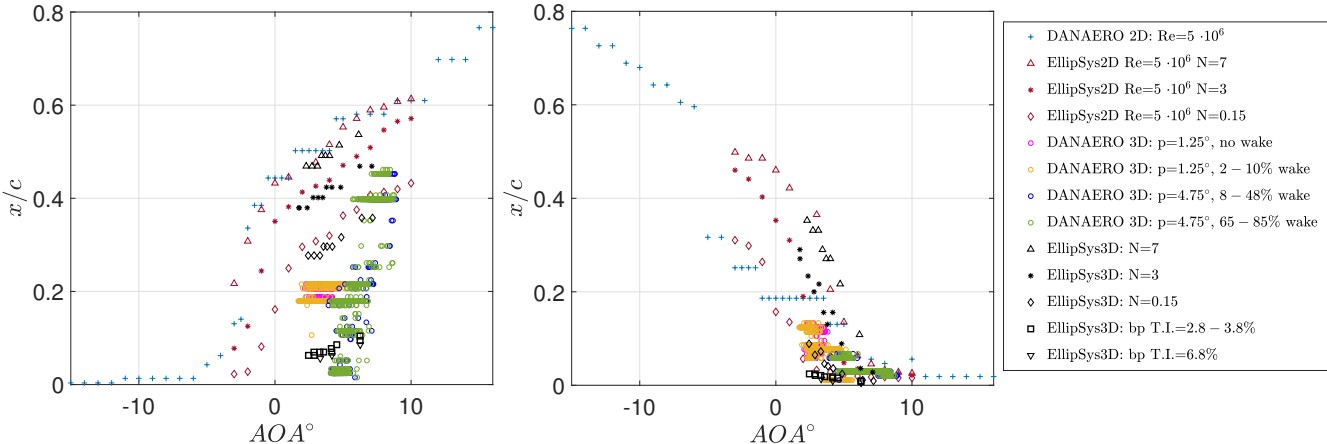

**Figure 14.** Detected transition points from the field and wind tunnel experiments and from the 2-D and 3-D CFD computations for the pressure side(left) and for the suction side(right).

pressure side between the 2D wind tunnel experimental results and 3D field experiments at several inflow conditions for the same manufactured airfoil geometry. For the high pitch case (p=4.75°), at AOA values from 4 to 9 degrees, the jump in the transition location seen in the 3D experiments is not visible either in wind tunnel results or in 2D or 3D simulations. These changes in the location of transition points during one revolution are represented by the simulations with different N factors for the natural transition model and different T.I. for the bypass transition model. The bypass transition model fits with the locations close to the leading edge and this shows that at several azimuthal positions, the transition is bypassed in the field experiments. The natural transition model with N=0.15 and N=3 fits to the positions where the transition locations are found to be at around $40\%$ of the chord. By combination of the results from the simulations with bypass and natural transition models, the scattered area from the experiments are covered. For the suction side (Figure 14 - right), the results from the 3D simulations with natural and bypass transition models cover most of the results from the field experiments. For the low AOA values, and low and no wake cases, the transition location movement is within $13\%$ along the chord in one revolution for the suction side. The more downstream transition locations, seen at low AOA values, fit with the results from the simulations with the natural transition model. This indicates that the natural transition type is also present on the suction side. Moreover, the 2D and 3D experimental results show agreement in the high AOA range on the suction side and the transition locations are in very close proximity of the leading edge in most of the cases.

## 5 Conclusions

In this study, the analysis of the field experiments and results from the 3D CFD simulations are presented to characterize the laminar-turbulent transition behaviour of a wind turbine under real atmospheric conditions. The data from high frequency



microphones placed on a wind turbine blade section are analyzed in the time and frequency domains. The transition locations are detected from the standard deviation of the pressure fluctuations, which are integrated between 2kHz and 7kHz. The inflow turbulence behaviour is obtained from one of the microphones placed nearby the leading edge by integrating the spectra from 100 to 300 Hz. The inflow velocity is obtained from meteorological mast measurements and used as an input parameter in the

CFD computations. The T.I. for the simulations is obtained from the relative velocity measurement from the pitot tube placed on the blade section.

The field experiment results showed that the transition behaviour on the wind turbine blade in real operating conditions differs from the model in the wind tunnel, caused by the influence of the inflow turbulence and the wake from another turbine. These factors change the relative velocity, so the effective AOA on the blade section, and, besides, inflow turbulence is observed

to have some direct effects on transition.

The effect of the wake is visible from the variation of the detected transition points at each revolution. As the wake affected rotor area increases, bigger jumps of the transition position is observed during one revolution. At the low and non-wake cases, each revolution is almost identical and the transition behaviour is mainly governed by the angle of attack changes due to the inflow velocity. The angle of attack effect on transition is analyzed by comparing results from the two different pitch settings

under similar inflow conditions. It is seen that for the pressure side, at low AOA cases, the transition position is not affected by the variations during a revolution as much as in the high AOA cases. Changes in AOA is found to be highly correlated with transition locations during a revolution and the variations among different revolutions are due to the inflow turbulence.

The normal sectional forces from the experiments and simulations are compared in order to quantify the rotational changes of the force, and analyze the differences among several revolutions. Moreover, by binning the sectional forces from the ex-

periments by the inflow velocity, the range that is covered by the simulation results obtained with various N numbers for the natural transition and T.I. for the bypass transition is identified. It is seen that the field experiments and the 3D simulations are comparable.

Furthermore, detected transition positions for the suction and pressure sides from the field experiments are compared with 3D CFD simulations. It is seen from the experiments that as the inflow fluctuations increase, the transition point moves closer

to the leading edge on the pressure side of the blade section. The EllipSys3D bypass transition model pressure side predictions are in good agreement with the experimental cases in conditions of high inflow turbulence at the azimuthal positions where the turbine is under the wake of an upstream turbine. On the other hand, the experimental result from the other azimuthal positions fit to the results obtained with the natural transition model. At these positions, the freestream velocities at different height levels of the ABL are matched with the azimuthal positions of the blade section. Comparing many datasets from different days, it is

seen that for high AOA and wake cases, the movement of the transition point covers up to $44\%$ of the chord on the pressure side in a single revolution, a value that drops to $5\%$ at low AOA and for no wake cases. On the suction side, changes in the transition position is also observable, and the field and wind tunnel experiments agree in the high AOA range. It is seen that, on the suction side, the effect of AOA is more prominent than the direct effect of the turbulence intensity, though it is not easy to reach a conclusion as the transition positions are in very close proximity to the leading edge (within $x/c = 1 - 13\%$).

Therefore, at these physical conditions, the suction side is not suited to distinguish the type of the transition mechanism. It is

visible from the pressure coefficient results that for azimuth angle 270°, where there is a wake from an upstream turbine for the presented case, the experiments fit with the low velocity 3D simulation results for natural and bypass transition models. On the other hand, 90° azimuthal position corresponds to high AOA region in the field experiments, a suction peak increase is observed and the EllipSys3D results that are simulated for the velocities during 10 minutes acquisition time fits with the experiments. For 0° azimuth, it is seen that 2D simulation and experimental results are within the standard deviation of 3D experiments. The pressure coefficient results also show the surface bumps on the suction side that could have been one of the factors effecting transition.

It is seen that the $e^N$ semi-empirical transition model and bypass transition model in EllipSys3D can be used for high Reynolds number flows (Re=5 million) in real atmospheric conditions. Using both models can cover the range of transition positions that is seen in the field experiments with a relevant choice of the amplification factors and T.I. values.

Several inflow scenarios are simulated separately in EllipSys3D as it is hard to control high turbulence in the wake region and handling varying N factor and T.I. in a single simulation. Simulations with more inflow characterization can be studied in the future in order to simulate the real inflow conditions from the experiments. Moreover, detailed characterization of the inflow turbulence measured on the blade with high sampling frequency instruments in field experiments is needed to separate relevant frequencies that affect boundary layer transition. By more field experiments and high resolution simulations, laminar-turbulent transition predictions can evolve, and eventually contribute to the aerodynamic prediction and the design of the wind turbine blades.

*Data availability.* Data is available upon request to corresponding author.

*Author contributions.* The literature review, the experimental data selection, processing and transition detection, the majority of the manuscript writing, mesh generation for EllipSys2D simulations, the EllipSys2D and 3D simulations and postprocessing of the simulations and comparisons of the results from the experiments and the simulations are conducted by Özçakmak. The DAN-AERO 3D and 2D experimental data and the guidance for the experimental data were provided by Madsen. The CFD mesh for the EllipSys 3D computations, and technical and theoretical guidance in CFD computations were provided by Sørensen N. N. The corrections in the manuscript and technical advice is provided by Sørensen J. N. All authors took part in writing and editing the paper.

*Competing interests.* The authors declare that they have no conflict of interest.



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
