# Peer review of "Laminar-turbulent transition characteristics of a 3-D wind turbine rotor blade based on experiments and computations"

_Wind Energy Science, 2020_

## Referee Comment (RC1) · Anonymous Referee #1 · 27 Mar 2020

Review of paper WES-2020-54 by

Özge Sinem Özçakmak, Helge Aagaard Madsen, Niels Nørmark Sørensen and Jens Nørkær Sørensen

With title

Laminar-turbulent transition characteristics of a 3-D wind turbine rotor blade based on experiments and computations

**General:**

The paper is about a comparison of results from the DAN-Aero measurement campaign with 3D-RANS-CFD regarding laminar to turbulent transition. This is important work, but the paper discusses many effects influencing the location of transition maybe not separated clearly enough to facilitate easy understanding and emphasising the main finding. The authors should think about "less is more".

**Specifics**

line 4: how does it benefit?

Lines 20/21: "Moreover, determining the relevant …" If you are able to answer this question, please state.

Line 28: are you able to quantify these differences?

Line 6 ff: Please explain, why you used this specific approach and it accuracy

Line 25: "identical" is impossible. Please state the accuracy (within xx micro-meters RMS or comparable)

Line 5: please state an equation how you calculated PSD from time series

Eq (1): "PSD" is not a suitable symbol. Use S or comparable.

Page 7:

Line 4: "transition locations" It seems that several different and not entirely equal definition of a "transition location" is used. Typically, you have a quantity which you relate to transition with a minimum (end of laminar part) and a following maximum (start of fully developed turbulent state). A lot of people take the maximum of slop in between as the "transition point". The authors should state if the use this terminology throughout the paper, and if it would not more consistent to talk about a "transition region".

Page 8:

First paragraph: please explain why you did nor made a mesh refinement study. 14 M cells seems to be very coarse.

Line 8: it may be helpful, to state the bypass mod el used already here.

Eq. please explain what the $z_i$ and $z_j$ are menaing

Line 9 "should be know …"  do you mean: are calculated to determine ?

Eqs 6 and 7: Cf -> $c_f$ ?

Eq 8: t_w -> \tau_w ?

Line 9/10: Pleas explain why you think that Mack's empirical relation is valid in these cases ?

Page 11:

Lines 2/3:  please state an equation, how TI enters here

Line 4. "0" is probably not possible (as it gives an N -> \infty). Please state the minimum N corresponding to TI = 2%.

In addition, a clear definition of TI (and the frequency range include) would be helpful before using this quantity.

Line 16/17: this   is not clear for me. How do get an intermittency factor from N (TS-scenario)?

Do you mean: the location closer/farer from the nose is used then ?

Lines 25 (and at other places in the text): State the difference between TI and "turbulence levels" and give an equation for the last one, if possible.

Line 30 ff: see above

First line: \gamma > 0.025. please give an explanation why this criteria is used and not \gamma = 0.5

(see my remarks above)

Line 20 ff

Please give reasons why 2 kHz is uses (why are frequency lower not important ?)

Line 22/23: I do not understand these two sentences at all. Please reformulate.

Is the sentence "Therefore, the PSD …) simply incomplete ?

Fig 4: the $L_p$ level are not equal (110 … 125, left and 65 … 115 right). Either adjust them or give reasons why this is not necessary.

Right: Please indicate Reynolds Number and AOA (ranges).

Line 14: Lp -> $L_p$ and Xtr -> ${x/c}_{tr}$ ?

Page 16:

Subsection title: Locations of laminar to turbulent calculated by CFD ?

Line 15: Pleas explain why N=3 was chosen

Page 17:

Line 3: FX -> F_x and FZ -> F_z ?

Page 18:

Fig 9:  To me this graph is overloaded. Obviously there two levels, so it seems to me if only the relevant CFD results related to those should be included. If possible reduce the number of data set in any case considerably.

Line 9/10 and

Fig 10:

Is it possible to increase line thickness with importance/degree of agreement?

By the way: Fig 10 to 12: Do you mean (x/c)_tr instead of Xtr ?

Page 21:

Fig 12: The acceptance and understanding of this graph would be greatly enlarged, if you  make the graph clearer: make the dots from CFD larger (on as a suggestion take the range (x/c)_tr_onset = gamma=0 to (x/c)_ft = gamma = 1 as an "error bar". Try to reduce to  measured point to value +/- std as well.

Line 11: typo "more more"

Page 23:

Fig 14: try to include a  fitted line (+/- std) for both CFD and measurements

Lines 2 to 4 (and earlier on several pages)

Try to correlate TI from  pure wind (measured in earth-fixed frame of reference) and "apparent" wind (measured in blade's rotating frame of reference)

Page 26 ff

References

Add:

doi :10.1088/1742-6596/1037/2/022012

Schaffarczyk et al.

Comparison of 3D transitional CFD simulations for rotating wind turbine wings with measurements

---

## Referee Comment (RC2) · Anonymous Referee #2 · 28 Apr 2020

The paper addresses the laminar-turbulent transition in boundary-layers over surface of wind turbine blades. The investigation includes analysis of field and wind tunnel measurements as well as evaluation of the transition prediction based on the commonly used methods.

The work is focused on effects of free-stream turbulence on laminar-turbulent transition and capability of empirical methods to predict it.

The topic of the work is of importance and necessary for correct prediction of the wind turbine energy production. The analysis of measured data is performed in an adequate manner trying to extract information as much as possible from the available data. Field

measurements of the kind reported here are difficult to perform and available data are usually limited.

I can recommend the paper for publication. However, there are some issues which should be addressed before the final acceptance:

1. Page 2, line 33: Bertolotti et al., J. Fluid Mech. (1992), vol. 242, pp. 441-474, is the first paper published on PSE, please use this instead of Herbert 1997.

2. Page 2, line 34-35: It is said 'the eN method is commonly used since it can predict the transition position accurately'. As authors mention later eN method requires calibration and cannot predict transition for a general flow condition. The correct sentence should be ' the eN method is commonly used since it can predict the trends in variation of transition position correctly'.

3. Page 3, line 15: The dimension of x-derivative of pressure level should be 'dB/m' instead of 'dB'.

4. Page 8, line 3: A reference to 'transient computation' is made. What is it meant with 'transient' here?

5. Page 9, line 5: Due to acceleration of flow, value of U varies in the wall-normal direction even outside of the boundary layer. How is the value of 'Ue' chosen at a given x/c location?

6. Page 9, line 10-11: How do the profiles obtained from von Karman boundary-layer equations compare to those given by CFD for cases studied here?

7. Page 9, line 24-25: Authors write 'In order to ensure that the entire stagnation line is found at each time step, Hiemenz flow across the stagnation line and Blasius flow along the stagnation line are assumed'. Please explain that further.

8. Page 11, line 4: It is said that eN method accurately predict transition for Tu up to 2%. One should be aware that the transition can be dominated by streak breakdown for

cases with Tu≈0.65%. See e.g. Suder, K., O'Brien, J. and Reshotko, E., Experimental study of bypass transition in a boundary layer. NASA TM 100913 (1988).

9. The pressure signals shown in figure 5a show a periodicity in x direction. Is there any physical explanation for that or it's just due to different sensitivity of microphones? It would be good to mark location of microphones in figure 4 and 5 and also to keep the same colorbar scaling in figures 5a and 5b.

10. Figure 9: I suggest to plot CFD data as vertical bars at the azimuthal positions corresponding to flow condition in simulation. I believe it will improve visualization of data. 11. Is there any flow conditions in present investigation at which transition is caused by flow separation?

12. Mention transition-prediction parameters for results in figure 13a-c. 13. Plots given in figure 14 are very difficult to read. Improv them by either dividing these plots to group of 2D relative 3D cases or using fix colors for 2D and 3D cases respectively.

14. Would be possible to perform a simulation allowing for modelling variation of FST level during one rotation by introducing time dependent transition N factor. Alternatively performing simple analysis by using for example Xfoil with input data (AoA, Re and FST level) from figure 6.
* * *

---

## Author Comment (AC1) · 31 Jul 2020

**Response to the Reviews**

We would like to thank the reviewers and the editor for their comments and improvement suggestions. We have tried to address all the concerns of the reviewers and think that the paper is improved with the suggestions. The comments of the reviewer are marked with *italic blue*.

Moreover, the modified parts of the text are highlighted with a turquoise color in the original manuscript.

Response to the RC 1:

> *Page 3*

1) *line 4: how does it benefit?*
   Solving laminar-turbulent transition with e^N method coupled with RANS provides the capability to discriminate the laminar and turbulent regions and it gives good agreements with experiments for high Reynolds number flows.
   The sentence in the manuscript is improved as:
   "The current analysis involves a coupling of the $e^N$ transition model with the RANS solver considering its accuracy for high Reynolds number flows in wind turbine applications"

2) *Lines 20/21: "Moreover, determining the relevant ..." If you are able to answer this question, please state.*
   For the field experiments, the high-frequency microphones close to the leading edge (in laminar boundary layer) are used to identify the frequency range for inflow turbulence. When we compare the PSD of the microphones in the laminar boundary layer between wind tunnel and rotor measurements, we see an increasing difference in the spectra from about 300 Hz and down to low frequencies, e.g. 30 Hz (see figure below [1]). Above 300 Hz the spectra are quite similar. When we then compare the field experiments with the wind tunnel results, we see a big difference in the transition positions. Therefore, we try to link these differences in transition to the differences in inflow turbulence represented by the integral of the spectra of the microphones in the laminar boundary layers is between 100 to 300 Hz. For the T-S wave frequency identification we have a previous study (Özçakmak, ÖS, Sørensen, NN, Madsen, HA, Sørensen, JN. Laminar-turbulent transition detection on airfoils by high-frequency microphone measurements. *Wind Energy*. 2019; 22: 1356– 1370. https://doi.org/10.1002/we.2361) for a wind tunnel case for various Reynolds numbers and AOA values, however, we haven't analyzed T-S wave frequencies in field experiments since it is unsteady, but this can be an important future work.
* * *
[1] Madsen, H. A., Özçakmak, Ö. S., Bak, C., Troldborg, N., Sørensen, N. N., and Sørensen, J. N.: Transition characteristics measured on a 2MW 80m diameter wind turbine rotor in comparison with transition data from wind tunnel measurements, https://doi.org/10.2514/6.2019-0801, https://arc.aiaa.org/doi/abs/10.2514/6.2019-0801, 2019a.

[Figure]

3) *Line 28: are you able to quantify these differences?*

In the literature, L/D ratio for fully turbulent and transitional flow on the same airfoil are compared. For example a paper from : 'Chaviaropoulos, P. K., Sieros, G., Prospathopoulos, J. M., Diakakis, K., & Voutsinas, S. G. Design and CFD-based Performance Verification of a Family of Low-Lift Airfoils.' , shows how can L/D ratio differ for 18% thick airfoil for a transitional and fully turbulent case that are used in design conditions.

[Figure]

Figure 7: Performance (L/D) of the 18% Low Lift 10-90 airfoil for transitional and fully turbulent flow conditions. Comparison among MaPFlow (CFD solver), Foil2w (viscous-inviscid interaction solver) and XFOIL calculations. Fixed transition locations were taken from XFOIL using the $e^N$ model with N=4

Moreover the low-speed shaft torque (LSSTQ) is also compared in a previous study for transitional and turbulent simulations and with measurements as seen below (Sørensen, N.N. (2009), CFD modelling of laminar-turbulent transition for airfoils and rotors using the γ − model. Wind Energ., 12: 715-733. doi:10.1002/we.325):

[Figure]

*The low-speed shaft torque of the NREL Phase VI rotor for fully turbulent and transitional conditions all compared to measured values*

A modification in the manuscript is done as follows:

"The difference between the design conditions for rotor and airfoils and the real operating conditions leads to inaccurate predictions of the loads and the performance as observed in previous studies \citep{nielscorrel} \citep{chaviaropoulosdesign}."

*Page 5*

4) *Line 6 ff: Please explain, why you used this specific approach and it accuracy*

Since the pitot tube was not working to measure the angle of attack at a specific blade section, we have chosen to use computations to find a correlation between the measured force and the angle of attack. We have observed a good correlation between computed and measured blade forces by the blade element momentum theory for attached flow. Refering to a Figure from IEA Task 29 Phase III (Boorsma K et al. Final report of IEA Wind Task 29 Mexnext (Phase 3), ECN-E--18-003 - January 2018. ):

[Figure]

It can be seen that a good correlation between all models for FN on the outboard part of the blade, (e.g. from radius 1.5 to 1.8m for e.g. 10m/s where the operational conditions are similar as in NM80 case). As the rotational speed for the NM80 case is constant in the case considered we have thus a direct link to AoA using a BEM model as in HAWC2 – blade element theory: Fn = ½ rho * Vrel^2 *Cl(alpha)*c Therefore, we have used the HAWC2 simulations to generate a correlation between AOA and the normal force and then this correlation is used with the measured forces to find the angle of attack. Based also on the many comparisons in the IEA Task 29 Phase II work we estimate the uncertainty on the AoA derivation to maximum +- 1 deg.

Moreover; a sentence is added to the manuscript: "A previous study has shown a good correlation between measured and computed normal forces \citep{boorsma2018final}."

5) *Line 25: "identical" is impossible. Please state the accuracy (within xx micro-meters RMS or comparable)*
The theoretical airfoil and the measured contour on the wind turbine blade is presented in an early work as follows:

[Figure]

As it can be seen there is a slight difference in the manufactured one. Then, this manufactured contour is used to manufacture an airfoil for the wind tunnel experiments.
 The manufacturing error is in few mms. However, it wasn't possible to reach this information since the experiments have been performed 10 years ago.

The sentence in the manuscript is modified as follows:
"The 2-D results shown in this paper originates from the wind tunnel experiments of the DAN-AERO project conducted in the LM wind tunnel with a reproduction of the actual blade section of the wind turbine deviating from the theoretical airfoil, details of which are explained in a previous study \citep{ozcakmakwj}."

6) *Line 5: please state an equation how you calculated PSD from time series*
This is done by short-time Fourier analysis. It is added to the text as giving a reference to the original paper for equations:
"For each time segment, the power spectral density (PSD) is \textcolor{csgr}{calculated by the short-time Fourier transformation analysis \citep{welch1967use}.}"

7) *Eq (1): "PSD" is not a suitable symbol. Use S or comparable.*

PSD is changed as $P_{s,p}$ as power spectral density of pressure fluctuations.

*Page 7:*

8) *Line 4: "transition locations" It seems that several different and not entirely equal definition of a "transition location" is used. Typically, you have a quantity which you relate to transition with a minimum (end of laminar part) and a following maximum (start of fully developed turbulent state). A lot of people take the maximum of slop in between as the "transition point". The authors should state if the use this terminology throughout the paper, and if it would not more consistent to talk about a "transition region".*

We define the transition position as the highest derivative of the RMS of the pressure fluctuations (Lp levels). The specific paragraph that says 'transition locations' as plural is because during a single revolution the transition location is detected at different chordwise positions due to angle of attack and inflow turbulence changes (turbine going in and out of the wake). The exact definition can be find in Page 6- Lines 14-16.

*Page 8:*

9) *First paragraph: please explain why you did nor made a mesh refinement study. 14 M cells seems to be very coarse.*

The problem is solved in 3 different grid levels in each case. As the convergence criteria is reached and the number of iterations are completed for the outer iterations, the solution is calculated in the next (finer grid level).

Moreover, the current investigation does not require a wake resolution. The error between the finest and a coarser mesh (with one-half number of cells in all directions) is within few percent's. Similar mesh is also used in a previous study: Madsen, M. H. A., Zahle, F., Sørensen, N. N., & Martins, J. R. (2019). Multipoint high-fidelity CFD-based aerodynamic shape optimization of a 10 MW wind turbine. *Wind Energy Science*, *4*(2), 163-192.)

The text is added in the manuscript as:  The mesh refinement study shows that the difference between the moment and forces for the finest and a coarser mesh (having one-half of the number of cells of the finest grid in all directions) is within a few percent as demonstrated in a previous study \citep{madsen2019multipoint}.

10) *Line 8: it may be helpful, to state the bypass model used already here.*

We would like to thank for the suggestion, we have applied it.

11) *Eq. please explain what the z_i and z_j are menaing*

They are several upstream and downstream locations from the rotor plane. Text added to the manuscript as follows:

"In order to calculate the induced axial velocity, $V_{ind}$, at the rotor plane, annular averaging of the axial velocity at several upstream and downstream locations (z_i and z_j) at a given radial location (in this case, the same blade section as in the microphone experiments) is performed."

*12) Line 9 "should be know …" do you mean: are calculated to determine?*

We have meant that they 'are needed as an input to determine….' Corrected like this in the text.

*13) Eqs 6 and 7: Cf -> c_f ?*

Corrected.

*14) Eq 8: t_w -> \tau_w ?*

Corrected.

*15) Line 9/10: Please explain why you think that Mack's empirical relation is valid in these cases?*

Mack's empirical relation is used an estimate in this case, it is also used inside EllipSys . Moreover, several N numbers are simulated in order to simulate range of occurrences.

*16) Lines 2/3: please state an equation, how TI enters here*

It enters as in 'Equation 12' indicated as Tu and it is defined in the text as the turbulence intensity at the transition onset. (Line 13 -14 - page 11)

Following text is modified as :

"\cite{abu1980natural} suggested that for the attached flows, transition onset can be obtained by correlating $Re_{\theta}$ to the free stream turbulence intensity as in Equation \ref{rettu}."

*17) Line 4. "0" is probably not possible (as it gives an N -> \infty). Please state the minimum N corresponding to TI = 2%. In addition, a clear definition of TI (and the frequency range include) would be helpful before using this quantity.*

The sentence is corrected in the manuscript as:

"….. predicts the transition for free stream turbulence levels , less than 2 \% for T-S dominated transition, but for higher levels it is bypassed.."

TI=2% corresponds to N value of 0.96. The definition for T.I is given at Page 12 Lines 3-4-5.

*18) Line 16/17: this is not clear for me. How do get an intermittency factor from N (TS-scenario)? Do you mean: the location closer/farer from the nose is used then?*

The x_{tr} in Equation 10 (Chen and Thyson formulation for intermittency) is determined by the amplification value (N) value, that is how N and intermittency is connected. Furthermore, a check is performed starting from the stagnation region and the criteria (for bypass or natural transition) that happens first in the flow direction is taken.

*19) Lines 25 (and at other places in the text): State the difference between TI and "turbulence levels" and give an equation for the last one, if possible.*

The places in the text are corrected and the T.I. calculated from the relative velocity measured by a pitot tube on the blade is separated from the turbulence levels ($L_{p,i}$) which is obtained from a leading edge microphone by integrating pressure fluctuations within the integral boundaries from 100 Hz to 300 Hz.

20) *Line 30 ff: see above*

Corrected in the text

21) *First line: \gamma > 0.025. please give an explanation why this criteria is used and not \gamma = 0.5 (see my remarks above)*

When we use *gamma > 0.025* for the transition criteria, considering our previous experiences, it shows a good correlation with XFOIL and a good agreement with experimental results, that is why this value is chosen.

22) *Line 20 ff Please give reasons why 2 kHz is uses (why are frequency lower not important ?)*

For the lower frequency range 2kHz is used. Frequencies lower than 2000 Hz are eliminated where the effect of inflow turbulence is dominant. Moreover, the 2kHz-7kHz range is also decided according the spectra where we observe biggest magnitude changes between laminar and turbulent spectra to detect transition more robust:

[Figure]

*Figure 2 PSD of the pressure fluctuations at Re= 3 million, AOA= 0 deg, suction side (integrated part highlighted with blue)*

We have also made a parametric study for this selection, it can be seen from the figure below with fm contours get closer forming a transition line at 2000-7000 Hz (d)

[Figure]

*Figure 3 Fm Contours for various frequency intervals at Re= 3 million, suction side*

a) 100-500 Hz

b) 700-1000 Hz

c) 1000-3000 Hz

d) 2000-7000 Hz

*23) Line 22/23: I do not understand these two sentences at all. Please reformulate. Is the sentence "Therefore, the PSD …) simply incomplete?*

Please see Figure 1 in this document. We see increasing differences in the spectra (laminar flow) from 300 Hz and down to very low frequencies. We have then chosen to use the integration of the spectra from 100-300 Hz to represent the inflow turbulence

The sentences are reformulated as:
"Several frequency intervals for PSD integration is also attempted in order to characterize the inflow turbulence from the microphone signals. It is seen in Figure \ref{fig:turbrange}-a that when the PSD is integrated in the frequency interval from 100 Hz to 300 Hz, the microphones closer to the leading edge shows high pressure levels on this frequency range capturing the pressure response to the inflow turbulence. Therefore, this frequency range is selected for the inflow analysis."

*Page 13*
*24) Fig 4: the L_p level are not equal (110 … 125, left and 65 … 115 right). Either adjust them or give reasons why this is not necessary.*

We would like to thank for the comment. Since the noise levels in the wind tunnel and the ones in the field experiments are different, for a clear representation of the both cases, the pressure levels (Lp) are presented in their own scale.

25) *Right: Please indicate Reynolds Number and AOA (ranges).*
A text is added to the Figure caption in the manuscript:
".... and for the blade section from the field experiments that corresponds to the Re=5.1 million and AOA varying from 3 to 8.5 degrees (b)."

26) *Line 14: Lp -> L_p and Xtr -> {x/c}_tr ?*
Corrected.

*Page 16:*
27) *Subsection title: Locations of laminar to turbulent calculated by CFD ?*
Title changed as: Laminar-turbulent transition locations calculated by CFD
28) *Line 15: Please explain why N=3 was chosen*
N=3 is used since it is within the scale and proximity of the measurements and usually gives the most aftward transition position on the pressure side.

*Page 17:*
29) *Line 3: FX -> F_x and FZ -> F_z ?*
Corrected.
*Page 18:*
30) *Fig 9: To me this graph is overloaded. Obviously there two levels, so it seems to me if only the relevant CFD results related to those should be included. If possible reduce the number of data set in any case considerably.*
The Figure is simplified with explanation and color blocks.
31) *Line 9/10 and* corrected
*Page 19*
32) *Fig 10: Is it possible to increase line thickness with importance/degree of agreement? By the way: Fig 10 to 12: Do you mean (x/c)_tr instead of Xtr ?*
The Xtr changed to x/c_tr. The figure is modified so that only degree of agreement parts are plotted and the results are grouped according to the N number of the simulations.

*Page 21:*
33) *Fig 12: The acceptance and understanding of this graph would be greatly enlarged, if you make the graph clearer: make the dots from CFD larger (on as a suggestion take the range (x/c)_tr_onset "error bar". = gamma=0 to (x/c)_ft = gamma = 1 as an Try to reduce to measured point to value +/- std as well.*
We would like to thank to the reviewer for the comments. The numerical data points are enlarged and the figure caption is modified. The standard deviation from the simulations is not possible in this case. However, the standard deviation from the experiments are studied for sectional force bins , however this also did not improve the graph representation. Therefore some visual improvements are applied.
*Page 22*
34) *Line 11: typo "more more"*
Corrected.
*Page 23:*

*35) Fig 14: try to include a fitted line (+/- std) for both CFD and measurements*
   The Figure is corrected by adding a fitted line to the CFD curves , by this way the representation of the plot is enhanced significantly. A text is added to the Figure caption as below:
   ( The EllipSys 2-D and 3-D results are presented as a fitted line of the data. Shadows around the lines show the standard deviation of the fit.).

*Page 24*
*36) Lines 2 to 4 (and earlier on several pages) Try to correlate TI from pure wind (measured in earth-fixed frame of reference) and "apparent" wind (measured in blade's rotating frame of reference)*
   We also would like see the suggested effect and the correlation; however the meteorological mast was not equipped with a fast-enough measurement device to characterize the turbulence at the scales that we are analyzing. Therefore, unfortunately it is not possible, but it is an important investigation for future experiments.

*Page 26 ff*
*37) References Add: doi :10.1088/1742-6596/1037/2/022012 Schaffarczyk et al.  Comparison of 3D transitional CFD simulations for rotating wind turbine wings with measurements*

The mentioned reference is added.

*Best Regards,*

Response to the RC 2:

*1. Page 2, line 33: Bertolotti et al., J. Fluid Mech. (1992), vol. 242, pp. 441-474, is the first paper published on PSE, please use this instead of Herbert 1997.*
The mentioned reference is added to the manuscript.

*2. Page 2, line 34-35: It is said 'the eN method is commonly used since it can predict the transition position accurately'. As authors mention later eN method requires calibration and cannot predict transition for a general flow condition. The correct sentence should be ' the eN method is commonly used since it can predict the trends in variation of transition position correctly'.*
We would like to thanks for this comment, the text is updated as : 'the e^N method is commonly used since it can predict the trends in variation of transition position accurately'.

*3. Page 3, line 15: The dimension of x-derivative of pressure level should be 'dB/m' instead of 'dB'.*

Since the derivation is performed for non-dimensional x/c, there is no meters in the units.

*4. Page 8, line 3: A reference to 'transient computation' is made. What is it meant with 'transient' here?*
The problem is solved in time. Although the parameters does not change in time, due to the high tip speed ratio of the problem, the transient calculations are used as a damping term by taking very small time steps for better convergence of the solution. 1200 steps per revolution is used in the current case as mentioned in the text.

Text is modified in the manuscript as follows:

"Due to the high tip speed ratio, in order to stabilize the simulations and enhance the convergence of the solution by small time steps, the simulations are performed by transient computations with 1200 steps per revolution."

*5. Page 9, line 5: Due to acceleration of flow, value of U varies in the wall-normal direction even outside of the boundary layer. How is the value of 'Ue' chosen at a given x/c location?*

We would like to thank for the comment. It is true and in the code, the tangential u velocity is calculated and the maximum tangential velocity is searched for. Then this maximum tangential velocity value is taken as the edge velocity.

It has also included in the manuscript as : "In EllipSys3D, the edge velocity ($U\_e$) is taken as the maximum tangential velocity."

*6. Page 9, line 10-11: How do the profiles obtained from von Karman boundary-layer equations compare to those given by CFD for cases studied here?*

In EllipSys, the e^N database is constructed based on the analytical boundary layer profiles. Therefore, in the computations, the boundary layer profile which is similar to the von Karman boundary layer profiles are identified. This is done to identify the velocity profiles and then the stability of the velocity profile is known. The edge velocities are taken with few other parameters and used in the boundary layer solver in the transition model. We take the information from the CFD solver and use it in a boundary layer solver inside this CFD solver. Internal tests has shown that the CFD profiles agrees quite well with the von Karman boundary layer profiles, but it is actually based on the boundary layer solution to avoid the need for excessively high discretization.

*7. Page 9, line 24-25: Authors write 'In order to ensure that the entire stagnation line is found at each time step, Hiemenz flow across the stagnation line and Blasius flow along the stagnation line are assumed'. Please explain that further.*

We use the Hiemenz flow to initiate the boundary layer solution. The location of the stagnation line is based on the several criterias such as pressure (the algorithm searches for a surface pressure (CP value of minus 1) and also check that if the velocity is at a divergence point. The algorithm looks at the stagnation pressure level, then it checks whether it is a cross flow or not. Hiemenz part is used only in order to get the initialization of the boundary layer parameters.

The text is modified in the manuscript as follows: "In order to ensure that the entire stagnation line is found at each time step, the pressure coefficient value is checked and the Navier-Stokes solution is analyzed. Along the stagnation line, Blasius flow and across the stagnation line Hiemenz flow are assumed."

*8. Page 11, line 4: It is said that eN method accurately predict transition for Tu up to 2%. One should be aware that the transition can be dominated by streak breakdown for cases with Tu_0.65%. See e.g. Suder, K., O'Brien, J. and Reshotko, E., Experimental study of bypass transition in a boundary layer. NASA TM 100913 (1988).*

We would like to thank to the reviewer for the comment. A text is added to the manuscript as follows:

"It should be also noted that transition can also be dominated by other mechanisms , for instance by a streak breakdown for turbulence levels around $0.65 \%$ \citep{suder1988experimental}"

*9. The pressure signals shown in figure 5a show a periodicity in x direction. Is there any physical explanation for that or it's just due to different sensitivity of microphones?*
We haven't observed any peaks in the spectra that shows periodicity. The observed case mentioned by the reviewer is due to the sensitivity of the microphones.

*It would be good to mark location of microphones in figure 4 and 5 and also to keep the same colorbar scaling in figures 5a and 5b.*
The colorbar scaling is corrected as suggested by the reviewer in Figure 5a and 5b.
The microphone placements on the pressure side are added to the Figure 4 as suggested. An explanation text is also added in the capture as:
"The airfoil with the microphone placement on the pressure side is presented on the left."

*10. Figure 9: I suggest to plot CFD data as vertical bars at the azimuthal positions corresponding to flow condition in simulation. I believe it will improve visualization of data.*
We would like to thank you for the suggestion. Figure 9 is updated for better representation of the results.

*11. Is there any flow conditions in present investigation at which transition is caused by flow separation?*
In the EllipSys code, the separation induced transition (bubble model) is also checked simultaneously with e^N transition model. Moreover, in the current experimental results presented here, we don't observe a flow separation.

*12. Mention transition-prediction parameters for results in figure 13a-c.*
Independent of the N number for the natural transition model , we obtain the same pressure coefficient value . And the amount of turbulence intensity that we are analyzing for the bypass transition model (2.8 to 6.8 %) gives also the same pressure coefficient value, therefore only one line is presented.

*13. Plots given in figure 14 are very difficult to read. Improve them by either dividing these plots to group of 2D relative 3D cases or using fix colors for 2D and 3D cases respectively.*

Figure 14 is improved by adding fitted lines and standard deviations as shadows for the CFD results.

*14. Would be possible to perform a simulation allowing for modelling variation of FST level during one rotation by introducing time dependent transition N factor. Alternatively performing simple analysis by using for example Xfoil with input data (AoA, Re and FST level) from figure 6.*

We would like to thank for the suggestions. It does not seem possible for us right now to obtain these results. This is the best we can do with available sources right now, it is not obvious for us to mimic this very complex scenarios.

Best Regards,

---

## Referee Report (RR1)

Review of WES-2020-54-V2

Small technical correction

Page 3: line 30: delete „." In the beginning